# A Novel Characterization of the Population Area Under the Risk Coverage Curve (AURC) and Rates of Finite Sample Estimators

Han Zhou [1]   Jordy Van Landeghem [2]   Teodora Popordanoska [1]   Matthew B. Blaschko [1]

## Abstract

The selective classifier (SC) has been proposed for rank based uncertainty thresholding, which could have applications in safety critical areas such as medical diagnostics, autonomous driving, and the justice system. The Area Under the Risk-Coverage Curve (AURC) has emerged as the foremost evaluation metric for assessing the performance of SC systems. In this work, we present a formal statistical formulation of population AURC, presenting an equivalent expression that can be interpreted as a reweighted risk function. Through Monte Carlo methods, we derive empirical AURC plug-in estimators for finite sample scenarios. The weight estimators associated with these plug-in estimators are shown to be consistent, with low bias and tightly bounded mean squared error (MSE). The plug-in estimators are proven to converge at a rate of $\mathcal{O}(\sqrt{\ln(n)/n})$ demonstrating statistical consistency. We empirically validate the effectiveness of our estimators through experiments across multiple datasets, model architectures, and confidence score functions (CSFs), demonstrating consistency and effectiveness in fine-tuning AURC performance.

## 1. Introduction

In safety-critical scenarios such as autonomous driving, medical diagnostics, and the justice system (Berk et al., 2021; Leibig et al., 2022; Dvijotham et al., 2023; Franc et al., 2023; Groh et al., 2024), selective classifiers (SC) are promising for their ability to withhold predictions under conditions of uncertainty, thereby mitigating associated risks and enhancing the reliability of the models (Geifman

and El-Yaniv, 2017; Geifman et al., 2019; Ding et al., 2020; Galil et al., 2023). Specifically, these classifiers employ cost-based models (Chow, 1970; Cortes et al., 2016; Hendrickx et al., 2024) with a reject region to balance the risks of wrong predictions against the non-decision costs. The goal of an effective SC system is to minimize the expected misclassification costs—termed *selective risk*—while maximizing *coverage*, ensuring the model provides accurate predictions for as many instances as possible. This dual focus on selective risk and coverage motivates the development and evaluation of SC systems.

Prominent evaluation metrics in SC systems, such as the area under the risk-coverage curve (AURC) and the normalized AURC (E-AURC) (Geifman et al., 2019), are widely used to assess model performance based on selective risk and coverage. While most studies interpret and improve (Ao et al., 2023; Traub et al., 2024) these metrics from the perspective of risk and coverage, relatively little attention has been given to directly optimizing SC models by treating AURC as a loss function. In addition, these metrics are typically computed empirically from the given datasets, making them susceptible to biases and variances, particularly in the context of a small finite sample rather than the underlying population. Franc et al. (2023) proposed the Selective Classifier Learning (SELE) loss as a lower bound of empirical AURC and is designed to optimize uncertainty scores by minimizing both the regression and SELE losses using batch training strategies. This approach only learns uncertainty scores on top of a pre-trained model within the selective classifier framework, and does not directly optimize the classifier itself based on the loss. Franc et al. (2023) motivate the SELE loss by the fact that it is "a close approximation of the AuRC and, at the same time, amenable to optimization." We demonstrate here through analysis of the computational complexity and statistical properties of direct AURC estimation, that approximation by a lower bound is unnecessary and both AURC and SELE are equally amenable to optimization.

We establish a formal definition of AURC at the population level based on the underlying data distribution and derive an equivalent expression that explicitly represents it as a reweighted risk function, where the weights are determined

[1]Processing Speech and Images, Department of Electrical Engineering, KU Leuven, Belgium [2]Instabase, San Francisco, USA. Correspondence to: Han Zhou <han.zhou@esat.kuleuven.be>.

*Proceedings of the $42^{nd}$ International Conference on Machine Learning*, Vancouver, Canada. PMLR 267, 2025. Copyright 2025 by the author(s).

solely by population rankings according to the CSFs. This formulation allows us to treat AURC as a loss function in a more theoretically grounded manner. Building upon these findings, we introduce two plug-in estimators with weight estimators derived from Monte Carlo method. We show that both can provide good estimation and come with theoretical guarantees. Specifically, we analyze the statistical properties of the weights estimators, including their MSE and bias, and establish the convergence rate of the plug-in estimators. Finally, we validate their efficacy through evaluations and fine-tuning experiments across various model architectures, CSFs, and datasets, demonstrating their practical advantages in AURC estimation.

## 2. Related Work

**Evaluation Metrics**: The Area Under the Risk Coverage curve (AURC) and its normalized counterpart Excess-AURC (E-AURC) (Geifman et al., 2019) are the most prevalent evaluation metrics for SC systems that compute the risk or the error with accepted predictions at different confidence thresholds. Furthermore, Cattelan and Silva (2024) have proposed a min-max scaled version of E-AURC, designed to maintain a monotonic relationship with AURC, thereby enhancing its consistency in performance assessment. However, Traub et al. (2024) argues that these metrics related to the selective risk, which only focus on the risk w.r.t. accepted predictions, do not suffice for a holistic assessment. To address this limitation, they developed the Area under the Generalized Risk Coverage curve (AUGRC), which quantifies the average risk of undetected failures across all predictions, thereby providing a comprehensive measure of system reliability. Despite these achievements, most studies directly employ the empirical AURC as a proxy for the population AURC, even in finite sample scenarios, without thoroughly examining the effectiveness of these estimators under such conditions. Franc et al. (2023) introduced the SELE score, a lower bound for AURC. However, their study did not explore important statistical properties of this estimator, such as its bias or MSE, when compared to the population AURC. In contrast to previous studies, our work focuses on formally defining the population AURC in terms of the underlying data distribution and offering a reliable approximation for it in finite sample settings. Our goal is to propose an effective estimator with theoretical guarantees and perform an empirical analysis to compare it with existing AURC estimators.

**Uncertainty estimation**: There has been a large number of works (Geifman et al., 2019; Abdar et al., 2021; Zhu et al., 2023) that highlight the importance of confidence scoring and uncertainty quantification associated with predictions. In practice[1], commonly used CSFs fall into two

main categories: ensemble approaches and post-hoc methods. Ensemble methods (Lakshminarayanan et al., 2017; Teye et al., 2018; Liu et al., 2023; Xia and Bouganis, 2023; Hou et al., 2023) require multiple forward passes to approximate the posterior predictive distribution, exemplified by Monte Carlo Dropout (MCD) techniques (Gal and Ghahramani, 2016). However, recent works (Cattelan and Silva, 2024; Xia and Bouganis, 2022) suggest that ensembles may not be crucial for enhancing uncertainty estimation but rather serve to improve the predictions through a set of diverse classifiers. Thus such methods are not considered further in this paper. In contrast, post-hoc estimators leverage the logits produced by the model to evaluate its performance. Popular methods include *Maximum Softmax Probability* (MSP) (Hendrycks and Gimpel, 2022), *Maximum Logit Score* (MaxLogit) (Hendrycks et al., 2022), *Softmax Margin* (Belghazi and Lopez-Paz, 2021), and *Negative Entropy* (Liu et al., 2020). Furthermore, Cattelan and Silva (2023) show that the maximum $p$-norm of the logits (MaxLogit-$p$Norm) can work better than MSP in uncertainty estimation when dealing with some models. Specifically, normalizing the logits with $\ell_2$ norm has been shown to yield more distinct confidence scores, as evidenced in (Wei et al., 2022). Gomes et al. (2022) propose the *Negative Gini Score*, which utilizes the squared $\ell_2$ norm of the softmax probability. In this study, we examine the impact of the post-hoc CSFs on the AURC, aiming to offer a thorough evaluation of AURC estimators in finite sample scenarios.

## 3. Performance Evaluation of Selective Classifiers

### 3.1. Problem Setting

Let $\mathcal{X} \subseteq \mathbb{R}^d$ be the input space, $\mathcal{Y} \subseteq \{0,1\}^k$ be the label space, and $P(x,y)$ be the unknown joint distribution over $\mathcal{X} \times \mathcal{Y}$. We consider a classifier $f : \mathcal{X} \to \Delta^k$, which maps to a $k$-dimensional probability simplex, and a *confidence scoring function* (CSF) $g : \mathcal{X} \to [0,1]$, the selective classification system $(f,g)$ at an input $x$ can then be described by

$$(f,g)(x) := \begin{cases} f(x) & \text{if } g(x) \geq \tau, \\ \text{``abstain''} & \text{otherwise.} \end{cases} \quad (1)$$

where "abstain" is triggered when $g(x)$ falls below a decision threshold $\tau \in \mathbb{R}$. Given a loss function $\ell : \Delta^k \times \mathcal{Y} \to \mathbb{R}$, the true risk of $f$ w.r.t. $P(x,y)$ is $R(f) = \mathbb{E}_{P(x,y)}[\ell(f(x),y)]$. Given the finite sample dataset $D_n = \{(x_i,y_i)\}_{i=1}^n \subseteq (\mathcal{X} \times \mathcal{Y})$ sampled i.i.d. from $P(x,y)$, the true risk can be inferred from the *empirical risk* $\hat{R}(f) := \frac{1}{n} \sum_{i=1}^n \ell(f(x_i),y_i)$. For practical purposes, we define the

---

[1]While it is preferable for the output domain of CSFs to be in $[0,1]$ for easier determination of selection thresholds, this is not a strict requirement.

selection function $\tilde{g}$ as $\tilde{g}(x) = \mathbb{I}[g(x) \geq \tau]$. The choice of $\tau$ depends on the specific scenario and the evaluation metric being used. It can either be a pre-defined constant or adapt dynamically based on the predicted uncertainty of the observations.

## 3.2. Evaluation Metrics

One common way to assess the performance of selective classifiers is the risk-coverage curve (RC curve) (El-Yaniv et al., 2010), where *coverage* measures the probability mass of the input space that is not rejected in Eq. (1), denoted by $\mathbb{E}_{P(x)}[\tilde{g}(x)]$. And the selective risk w.r.t. $P(x, y)$ is then defined as

$$R(f, \tilde{g}) := \frac{\mathbb{E}_{P(x,y)}[\ell(f(x), y)\tilde{g}(x)]}{\mathbb{E}_{P(x)}[\tilde{g}(x)]}. \quad (2)$$

$\ell$ is typically the 0/1 error, making $R(f, \tilde{g})$ the selective error. As indicated by the equation above, risk and coverage are strongly dependent, where rejecting more examples reduces selective risk but also results in lower coverage. This relationship revealed by the curve motivates the development of more nuanced evaluation metrics for selective classifiers. Additionally, accuracy alone often fails in cases of class imbalance or pixel-level tasks (Ding et al., 2020), so evaluation metrics should accommodate different loss functions for a more comprehensive assessment.

## 3.3. Equivalent Expressions of AURC

Driven by the aforementioned considerations, the AURC metric (Geifman et al., 2019) is designed to offer a robust evaluation framework for classifiers by effectively capturing performance across varying rejection thresholds that are determined based on the distribution of samples within the population. The AURC is typically specified as an empirical quantity from a finite sample (Franc et al., 2023, Eq. (27)), from which we derive the population AURC as

$$\text{AURC}_p(f) = \mathbb{E}_{\tilde{x} \sim P(x)} \frac{\mathbb{E}_{(x,y) \sim P(x,y)}\ell(f(x), y))\mathbb{I}[g(x) \geq g(\tilde{x})]}{\mathbb{E}_{x' \sim P(x)}\mathbb{I}[g(x') \geq g(\tilde{x})]}. \quad (3)$$

Noticing that the expectation in the numerator can be swapped with the expectation outside, the equation above can then be written as:

$$\text{AURC}_p(f) = \mathbb{E}_{(x,y) \sim P(x,y)}[\alpha(x)\ell(f(x), y)] \quad (4)$$

where

$$\alpha(x) = \mathbb{E}_{\tilde{x} \sim P(x)}\left(\frac{\mathbb{I}[g(x) \geq g(\tilde{x})]}{\mathbb{E}_{x' \sim P(x)}\mathbb{I}[g(x') \geq g(\tilde{x})]}\right) \quad (5)$$

This expression shows that the population AURC can be interpreted as the expectation of the risk function weighted by $\alpha(x)$ that accounts for the importance of each point. In

order to better understand the population AURC, we study the behavior of the weight $\alpha(x)$ in Eq. (5). The following proposition provides an equivalent expression for $\alpha(x)$.

**Proposition 3.1** (An equivalent expression of $\alpha(x)$)**.** *Define function $G(x)$ as the cumulative distribution function(CDF) of the CSF $g(x)$ such that*

$$G(x) = Pr(g(x') \leq g(x)) = \int \mathbb{I}[g(x') \leq g(x)] dP(x'). \quad (6)$$

*Under this definition, the $\alpha(x)$ in Eq. (5) is equivalent to*

$$\alpha(x) = -\ln(1 - G(x)). \quad (7)$$

*Proof.* Since the expectation in the denominator in Eq. (5) is the CDF of $1 - G(x)$, we have:

$$\alpha(x) = \int_{\tilde{x}} \frac{\mathbb{I}[g(x) \geq g(\tilde{x})]}{1 - G(\tilde{x})} dP(\tilde{x}).$$

This implies that we are integrating over the domain $\tilde{x}$ where $g(\tilde{x}) \leq g(x)$. Hence, we can rewrite it as:

$$\alpha(x) = \int_{g(\tilde{x}) \leq g(x)} \frac{1}{1 - G(\tilde{x})} dP(\tilde{x}).$$

To proceed, note that $G(\tilde{x})$ is the CDF of $g(\tilde{x})$, and since $G(\tilde{x})$ is monotonically increasing in $g(\tilde{x})$, we can reparameterize the integral in terms of $G(\tilde{x})$. Specifically, we know that $G(\tilde{x})$ takes values between $0$ and $1$ as it is a CDF, thus integration can be rewritten as:

$$\alpha(x) = \int_{G(\tilde{x}) \leq G(x)} \frac{1}{1 - G(\tilde{x})} dP(\tilde{x}) = \int_0^{G(x)} \frac{1}{1 - G(\tilde{x})} dP(\tilde{x}).$$

Now, this integral is straightforward to compute:

$$\int_0^{G(x)} \frac{1}{1 - t} dt = -\ln(1 - G(x)).$$

Thus, we have derived the desired result:

$$\alpha(x) = -\ln(1 - G(x)).$$

$\square$

Here $G(x)$ can be interpreted as the population rank percentile based on the CSF sorted in ascending order. This proposition motivates the following formulation, which can be considered equivalent to the population AURC in Eq. (3).

**Definition 3.2** (An equivalent expression of $\text{AURC}_p$)**.** Given $G(x)$ as the CDF of the random variable $g(x)$, the population AURC in Eq. (3) is equivalent to:

$$\text{AURC}_a(f) = \int \alpha(x)\ell(f(x), y)dP(x, y) \quad (8)$$

where $\alpha(x) = -\ln(1 - G(x))$.

We also provide empirical evidence supporting the equivalence in Appendix A.5. Notably, the following integral holds:

$$\int_0^1 -\ln(1-z)\,dz = 1. \tag{9}$$

This result indicates that, in the limit of infinite data, the integral of $\alpha(x)$ computed using this formula converges to one. Consequently, the population AURC can be interpreted as a redistribution of the risk.

### 3.4. Plug-in Estimator of AURC

Given a finite sample of size $n$, namely $D_n = \{(x_i, y_i)\}_{i=1}^n \subseteq (\mathcal{X} \times \mathcal{Y})$, which is sampled i.i.d. from the joint probability distribution $P(x,y)$, following Eq. (3), the empirical AURC can be defined based on the sample as

$$\widehat{\text{AURC}}_p(f) = \frac{1}{n}\sum_{j=1}^n \frac{\frac{1}{n}\sum_{i=1}^n \ell(f(x_i), y_i)\mathbb{I}[g(x_i) \geq g(x_j)]}{\frac{1}{n}\sum_{k=1}^n \mathbb{I}[g(x_k) \geq g(x_j)]}. \tag{10}$$

This formulation represents the widely used AURC metric for evaluating the SC system. However, guarantees on the relationship to population-level AURC has not been considered, even though relying on this empirical estimator may introduce error, particularly when assessing the SC system with a small sample size. The naive implementation of this estimator incurs a quadratic computational cost of $\mathcal{O}(n^2)$ due to the nested loops. However, some packages e.g. *torch-uncertainty*[2] decrease this complexity to $\mathcal{O}(n\ln(n))$ by replacing redundant subset evaluations with a single sorting step followed by cumulative summation that efficiently computes error rates across all coverage levels. Here, we present a derivation of a method that achieves a computational complexity of $\mathcal{O}(n\ln(n))$. By leveraging the approach used to transform Eq.(3) into Eq.(4), the empirical AURC can be reformulated as a plug-in estimator:

$$\widehat{\text{AURC}}_p(f) = \frac{1}{n}\sum_{i=1}^n \hat{\alpha}(x_i)\ell(f(x_i), y_i) \tag{11}$$

where

$$\hat{\alpha}(x_i) = \sum_{j=1}^n \frac{\mathbb{I}[g(x_i) \geq g(x_j)]}{\frac{1}{n}\sum_{k=1}^n \mathbb{I}[g(x_k) \geq g(x_j)]} = \sum_{j=1}^{r_i} \frac{1}{n-j+1} \tag{12}$$

where $r_i$ denotes the rank of $x_i$ when the data is sorted in ascending order according to the CSF, such that a larger $r_i$ corresponds to a higher CSF value. For simplicity, we use $\hat{\alpha}_i$ as shorthand for $\hat{\alpha}(x_i)$. This estimator is a consistent estimator of $\alpha_i$ for the population AURC, as directly established by the

[2]https://github.com/ENSTA-U2IS-AI/torch-uncertainty

Continuous Mapping Theorem. Let $\text{H}_n := \sum_{k=1}^n \frac{1}{k}$ denote the $n$th harmonic number, and define the digamma function as $\psi(n) := \frac{\Gamma'(n)}{\Gamma(n)}$. The relationship between these two functions is given by $\text{H}_n = \psi(n+1) + \gamma$, where $\gamma \approx 0.577$ is the Euler–Mascheroni constant. Setting $\text{H}_0 = 0$, we can express $\hat{\alpha}_i$ in terms of harmonic numbers or digamma functions:

$$\hat{\alpha}_i = \text{H}_n - \text{H}_{n-r_i} = \psi(n+1) - \psi(n-r_i+1), \tag{13}$$

which enables efficient computation of the weight estimator. The computation cost of Eq. (11) is $\mathcal{O}(n\ln(n))$ due to the sorting operation required for rank computation. Additionally, for the finite sample case, we have

$$\frac{1}{n}\sum_{i=1}^n \hat{\alpha}_i = \frac{1}{n}\sum_{i=1}^n (H_n - H_{n-i}) = 1, \tag{14}$$

indicating the plug-in estimator with this weight $\hat{\alpha}_i$ can be viewed as a redistribution of the risk. Each individual loss is weighted by $\frac{1}{n}\hat{\alpha}_i$, which depends on the rank of the corresponding sample point.

Franc et al. (2023) gave another expression of the empirical AURC in Eq. (10) that can be interpreted as an arithmetic mean of the empirical selective risks corresponding to the coverage spread evenly over the interval $[0,1]$ with step $\frac{1}{n}$. In addition, they proposed the SELE score, served as a coarse lower bound for empirical AURC. Details about this estimator are provided in Appendix A.2.

### 3.5. Alternate Derivation of Plug-in Estimators via Monte Carlo

In this section, we explore an alternative derivation of plug-in estimators using the Monte Carlo method. For our population $\text{AURC}_a$ in Eq. (8), we aim to estimate this quantity using Monte Carlo integration. Since the cumulative distribution score $G(x)$ is unknown, we require an estimator for Eq. (7), which can be achieved by taking the conditional expectation

$$\mathbb{E}\left[-\ln(1-G(x_i))|\{g(x_j)\}_{1\leq j\leq n}\right]. \tag{15}$$

Since $\{G(x_i)\}_{1\leq i\leq n}$ behave as i.i.d. samples from a uniform distribution $\mathcal{U}[0,1]$ when $x_i$ are i.i.d. from $P(x)$, we sample i.i.d. $\{\beta_i\}_{1\leq i\leq n} \sim \mathcal{U}[0,1]$, then sort this set in ascending order. Let $r_i$ be the rank for $\beta_i$ and set $\alpha_i = -\ln(1-\beta_i)$. Consequently, we can find a lower variance estimate with the same bias by repeating the process and averaging the obtained $\alpha_i$ estimates. The $\beta_i$ are order statistics of the uniform distribution, and have known distribution (Jones, 2009, Section 2)

$$\beta_i \sim \text{Beta}(r_i, n+1-r_i) = \frac{\beta_i^{r_i-1}(1-\beta_i)^{n-r_i}}{\text{B}(r_i, n+1-r_i)} \tag{16}$$

where B denotes the beta function. Consequently, the limit expectation of our estimator with repeatedly resampled $\beta_i$ will yield

$$
\begin{aligned}
\hat{\alpha}_i &= \mathbb{E}_{\beta_i}[-\ln(1 - \beta_i)] \\
&= \int_0^1 -\ln(1 - x)\frac{x^{r_i - 1}(1 - x)^{n - r_i}}{B(r_i, n + 1 - r_i)}dx \quad (17) \\
&= H_n - H_{n - r_i}
\end{aligned}
$$

which leads to the weight estimator in Eq. (13). This indicates that the plug-in estimator of Sec. 3.4 is in fact quite principled. Furthermore, we demonstrate the consistency of this weight estimator in Prop. A.4 and 3.6. In the above procedure, we have used $\hat{\alpha}_i = \mathbb{E}_{\beta_i}[-\ln(1 - \beta_i)]$, but we can utilize the expectation of $\beta_i$, leading to another weight estimator

$$
\hat{\alpha}_i' = -\ln(1 - \mathbb{E}_{\beta_i}[\beta_i]) = -\ln\left(1 - \frac{r_i}{n + 1}\right) \quad (18)
$$

where the last equation is due to the fact that the expectation of $\text{Beta}(a, b)$ is $\frac{a}{a+b}$. This estimator is consistent, as $\lim_{n \to \infty}\left(\frac{r_i}{n + 1}\right) = \beta_i$. In addition, we have

$$
\frac{1}{n}\sum_{i=1}^n \hat{\alpha}_i' = -\frac{1}{n}\sum_{i=1}^n \ln\left(1 - \frac{r_i}{n + 1}\right) < 1, \quad (19)
$$

but it approaches one as $n \to \infty$. Since the function $m(t) = -\ln(1 - t)$ is convex, applying Jensen's inequality gives:

$$
-\ln(1 - \mathbb{E}_{\beta_i}[\beta_i]) \le \mathbb{E}_{\beta_i}[-\ln(1 - \beta_i)], \quad (20)
$$

indicating the first estimator $\hat{\alpha}_i$ upper bounds the $\hat{\alpha}_i'$. Thus this weight estimator $\hat{\alpha}_i'$ will lead to a consistent plug-in estimator that lower bounds the plug-in estimator with $\hat{\alpha}_i$. In the next section, we will analyze the statistical properties of these two plug-in estimators, incorporating the weight estimators discussed above.

### 3.6. Statistical Properties

In this section, we show that empirical estimators of AURC are in general biased (Prop. 3.3, 3.4), but we also show consistency at a favorable convergence rate (Prop. 3.6) indicating the soundness of empirical estimators even at relatively small batch sizes. Given the fact that the weight estimator and the losses are typically not independent as they both depend on the model logits, it is difficult to directly derive the statistical properties of the plug-in estimators. Therefore, we begin by examining the properties of the weight estimators $\hat{\alpha}_i$ and $\hat{\alpha}_i'$ based on finite samples. For a specific data pair $(x_i, y_i) \in (\mathcal{X}, \mathcal{Y})$ with an unknown population rank percentile $\beta_i$, we consider randomly sampling $n - 1$ i.i.d. samples repeatedly from the population. Our analysis focuses on assessing the bias and MSE of the weight estimators associated with this data pair.

**Proposition 3.3** (**Bias of $\hat{\alpha}_i$**). *The bias of the $\hat{\alpha}_i$ is given by*

$$
\begin{aligned}
&\text{Bias}(\hat{\alpha}_i | G(x_i) = \beta_i) \quad (21) \\
&= H_n - \sum_{i=1}^n H_{n-i}C_{n-1}^{i-1}\beta_i^{i-1}(1 - \beta_i)^{n-i} + \ln(1 - \beta_i).
\end{aligned}
$$

*Proof.* The conditional expectation of $\hat{\alpha}_i$ associated with $(x_i, y_i)$ is given by

$$
\mathbb{E}[\hat{\alpha}_i | G(x_i) = \beta_i] = \sum_{i=1}^n (H_n - H_{n-i})\Pr(r_i = i | G(x_i) = \beta_i).
$$

We notice that $\Pr(r_i | G(x_i) = \beta_i)$ is a binomial distribution $\text{Bin}(n - 1, \beta_i)$ given by:

$$
\Pr(r_i = i | G(x_i) = \beta_i) = C_{n-1}^{i-1}\beta_i^{i-1}(1 - \beta_i)^{n-i}
$$

because the probability of any i.i.d. sample being ranked below $x_i$ is $\beta_i$, and there must be $(i - 1)$ of them for the sample to be ranked $i$th, while the remaining $(n - i)$ samples must be ranked higher, each with an independent probability of $(1 - \beta_i)$. Combining these results gives us:

$$
\begin{aligned}
&\text{Bias}(\hat{\alpha}_i | G(x_i) = \beta_i) = \mathbb{E}[\hat{\alpha}_i | G(x_i) = \beta_i] - \alpha_i \\
&= H_n - \sum_{i=1}^n H_{n-i}C_{n-1}^{i-1}\beta_i^{i-1}(1 - \beta_i)^{n-i} + \ln(1 - \beta_i).
\end{aligned}
$$

which concludes our proof. $\square$

**Proposition 3.4** (**Bias of $\hat{\alpha}_i'$**). *The bias of the $\hat{\alpha}_i'$ is given by*

$$
\begin{aligned}
&\text{Bias}(\hat{\alpha}_i' | G(x_i) = \beta_i) \quad (22) \\
&= -\sum_{i=1}^n \ln\left(1 - \frac{i}{n + 1}\right)C_{n-1}^{i-1}\beta_i^{i-1}(1 - \beta_i)^{n-i} + \ln(1 - \beta_i).
\end{aligned}
$$

*Proof.* The expected $\hat{\alpha}_i'$ associated with $(x_i, y_i)$ is given by

$$
\begin{aligned}
&\mathbb{E}[\hat{\alpha}_i' | G(x_i) = \beta_i] \\
&= -\sum_{i=1}^n \ln\left(1 - \frac{i}{n + 1}\right)\Pr(r_i = i | G(x_i) = \beta_i).
\end{aligned}
$$

Since $\Pr(r_i | G(x_i) = \beta_i)$ follows $\text{Bin}(n - 1, \beta_i)$, we obtain:

$$
\begin{aligned}
&\text{Bias}(\hat{\alpha}_i' | G(x_i) = \beta_i) = \mathbb{E}[\hat{\alpha}_i' | G(x_i) = \beta_i] - \alpha_i \\
&= -\sum_{i=1}^n \ln\left(1 - \frac{i}{n + 1}\right)C_{n-1}^{i-1}\beta_i^{i-1}(1 - \beta_i)^{n-i} + \ln(1 - \beta_i)
\end{aligned}
$$

which concludes our proof. $\square$

For the above mentioned weight estimators, the SELE Weight estimator $\hat{\alpha}_i^{se}$ exhibits the largest bias compared with $\hat{\alpha}$ and $\hat{\alpha}'$, as indicated in Fig 1. Due to this significant bias in weight estimation, SELE is not a reliable estimator for population AURC.

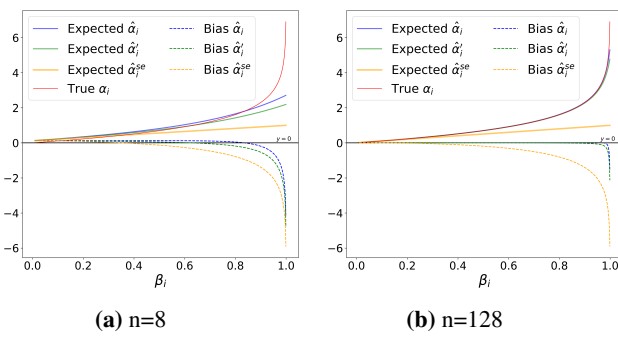

**(a)** n=8        **(b)** n=128

*Figure 1.* The bias of the weights estimator as a function of $\beta$, based on the results in Propositions 3.3, 3.4 and A.2. The bias of these weight estimators is not equal to zero in general, but being positive for smaller $\beta_i$ and negative for larger $\beta_i$ as indicated. As sample size increases, the expected bias decreases significantly for larger $\beta_i$. The complete plots for different sample size can be found in Appendix S8.

**Proposition 3.5** (**MSE of $\hat{\alpha}_i$**). *The MSE of the $\hat{\alpha}_i$ is*

$$\mathrm{MSE}(\hat{\alpha}_i) = \psi'(n+1-r_i) - \psi'(n+1)$$

$$\asymp \frac{\beta_i}{n(1-\beta_i)+1}. \tag{23}$$

*Proof.* From result in Prop.3.1, we calculate:

$$\mathrm{MSE}(\hat{\alpha}_i) = \mathbb{E}_{\beta_i}\left[(\hat{\alpha}_i + \ln(1-\beta_i))^2\right]$$

$$= \int_0^1 \left((\mathrm{H}_n - \mathrm{H}_{n-r_i}) + \ln(1-\beta_i)\right)^2 d\mathrm{P}(\beta_i)$$

$$= \underbrace{\int_0^1 \ln(1-\beta_i)^2 d\mathrm{P}(\beta_i) - (\mathrm{H}_n - \mathrm{H}_{n-r_i})^2}_{:=M}$$

where the second equality is led by $\int_0^1 \ln(1-\beta_i)d\mathrm{P}(\beta_i) = -(\mathrm{H}_n - \mathrm{H}_{n-r_i})$. And $d\mathrm{P}(\beta_i)$ is taken to mean integration with respect to the measure induced by $\beta_i \sim \mathrm{Beta}(r_i, n+1-r_i)$. Focusing on the remaining integral, we have the closed form:

$$M = (\mathrm{H}_n - \mathrm{H}_{n-r_i})^2 + \psi'(n+1-r_i) - \psi'(n+1),$$

and the result:

$$\mathrm{MSE}(\hat{\alpha}_i) = \psi'(n+1-r_i) - \psi'(n+1).$$

This term involves the first derivative of the digamma function for which the inequality $\frac{1}{n} + \frac{1}{2n^2} \leq \psi'(n) \leq \frac{1}{n} + \frac{1}{n^2}$ is well known. Applying these inequalities, we obtain

$$\mathrm{MSE}(\hat{\alpha}_i) \leq \frac{1}{n+1-r_i} + \frac{1}{(n+1-r_i)^2} - \frac{1}{n+1} - \frac{1/2}{(n+1)^2}$$

$$= \mathcal{O}\left(\frac{1}{n-r_i+1} - \frac{1}{n+1}\right) = \mathcal{O}\left(\frac{\beta_i}{n(1-\beta_i)+1}\right).$$

By analogous reasoning determining a lower bound on the MSE, we achieve the result

$$\mathrm{MSE}(\hat{\alpha}_i) \asymp \frac{\beta_i}{n(1-\beta_i)+1} \quad \forall \beta_i \in (0,1). \tag{24}$$

which is visualized in Fig 2. $\qquad\qquad\square$

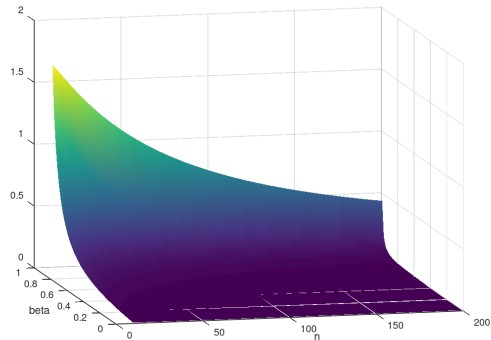

*Figure 2.* The bound in Eq. (24) as a function of $n$ and $\beta_i$.

We also demonstrate in Appendix A.3 that the MSE of $\hat{\alpha}_i'$ is tightly upper bounded by Eq. (24), though it remains larger than the MSE of $\hat{\alpha}_i$.

**Proposition 3.6** (**Convergence Rate of the plug-in estimators with $\hat{\alpha}_i$ or $\hat{\alpha}_i'$**). *Assume that the loss function $\ell$ is square-integrable, i.e., $\int \ell^2(f(x), y) \, dP(x,y) < \infty$. Then, the plug-in estimators with $\hat{\alpha}_i$ or $\hat{\alpha}_i'$ as the weight estimator, converges at a rate of $\mathcal{O}(\sqrt{\ln(n)/n})$.*

*Proof.* We first analyze the difference between the plug-in estimator with $\hat{\alpha}_i$ and the population expected value:

$$\frac{1}{n}\sum_{i=1}^n \hat{\alpha}_i \ell(f(x_i), y_i) - \mathbb{E}[\alpha \ell(f(x), y)].$$

This can be decomposed as the sum of the following two terms:

$$A = \frac{1}{n}\sum_{i=1}^n (\hat{\alpha}_i - \alpha_i)\ell(f(x_i), y_i)$$

$$B = \frac{1}{n}\sum_{i=1}^n \alpha_i \ell(f(x_i), y_i) - \mathbb{E}[\alpha \ell(f(x), y)]$$

where term (1) captures the error caused by the bias in estimating $\alpha_i$ and term (2) represents the error introduced by approximating the expected value $\mathbb{E}[\alpha \ell(f(x), y)]$ with the empirical average.

Making use of the Cauchy–Schwarz inequality, we obtain

the following result:

$$A^2 \leq \left( \frac{1}{n} \sum_{i=1}^{n} (\hat{\alpha}_i - \alpha_i)^2 \right) \left( \frac{1}{n} \sum_{i=1}^{n} \ell(f(x_i), y_i)^2 \right)$$

$$= \left( \frac{1}{n} \sum_{i=1}^{n} (\hat{\alpha}_i - \alpha_i)^2 \right) \mathbb{E} \left[ \ell(f(x), y)^2 \right]. \quad (25)$$

From Proposition A.4, $\frac{1}{n} \sum_{i=1}^{n} \mathrm{MSE}(\hat{\alpha}_i)$ is bounded by $\mathcal{O}\left( \frac{\ln(n)}{n} \right)$, which means

$$\frac{1}{n} \sum_{i=1}^{n} (\hat{\alpha}_i - \alpha_i)^2 = \mathcal{O}\left( \frac{\ln(n)}{n} \right). \quad (26)$$

By combining this with Eq. (25) and the square-integrable assumption of the loss function $\ell$, term A asymptotically converges at a rate $\mathcal{O}(\sqrt{\ln(n)/n})$. Term B, corresponding to the Monte Carlo method, is well-known to converge at a rate $\mathcal{O}(n^{-1/2})$ (Caflisch, 1998), which is faster than $\mathcal{O}(\sqrt{\ln(n)/n})$. Thus, our overall convergence rate is dominated by the rate derived for term A. Similarly, the same convergence rate applies to the estimator with $\hat{\alpha}_i'$. □

# 4. Experiments

**Datasets**. We use images datasets such as CI-FAR10/100 (Krizhevsky et al., 2009) and ImageNet (Deng et al., 2009), and a text dataset i.e Amazon Reviews (Ni et al., 2019). The Amazon dataset contains review text inputs paired with 1-out-of-5 star ratings as labels.

**Models**. For experiments on CIFAR10/100, we report the results on the VGG13, VGG16, VGG19 (Simonyan and Zisserman, 2014) model with batch norm layers, WideResNet28x10 (Zagoruyko and Komodakis, 2016), and ResNet (He et al., 2016) models with different depths $(20, 56, 110, 164)$. For each model architecture, we have 5 different models that are pre-trained on the CIFAR10/100 dataset. For experiments on Amazon dataset, we use pre-trained transformer-based models – BERT (Kenton and Toutanova, 2019), RoBERTa (Liu et al., 2021a), Distill-Bert (Sanh, 2019) (D-BERT), and Distill-Roberta (D-RoBERTa)[3]. For experiments on the ImageNet dataset, we use the pre-trained models from *timm* (Wightman, 2019) package, including two vision transformer (ViT) (Dosovitskiy et al., 2020) variants, ViT-Small and ViT-Large; and two Swin transformer-based models (Liu et al., 2021b), Swin-Base and Swin-Tiny. All these models are configured with standard image resolution – 224.

**Metrics.** For our comparative analysis, we evaluate several metrics, including the population $\mathrm{AURC}_p$ and finite-sample estimators. The $\mathrm{AURC}_p$ is computed using Eq. (11) across

---

[3]Obtained from RoBERTa with the procedure of Sanh (2019)

the test set. In the finite-sample setting, we evaluate the plug-in estimators with $\hat{\alpha}$ or $\hat{\alpha}'$, the SELE score (Franc et al., 2023), and $2 \times$SELE as proposed by the original authors (see Appendix A.2 for a discussion). Beyond the 0/1 loss, we incorporate these metrics with the Cross-Entropy (CE) loss that serves as a complementary measure for assessing the classifier's performance.

**Experimental setup.** We evaluate the metrics using several pre-trained models on the test set, which is randomly divided into various batch sizes $(8, 16, 32, \cdots, 1024)$. We use MSP as our confidence score function, compute the metrics for these batch samples, and subsequently calculate the mean and standard deviation of these finite sample estimators. The population $\mathrm{AURC}_p$ is computed across all samples in the test set. For the CIFAR10/100 datasets, we evaluate the mean and standard deviation of the Mean Absolute Error (MAE) across five distinct pre-trained models. For the ImageNet and Amazon datasets, we compute the mean, standard deviation, and MSE for different estimators of the pre-trained model across batch samples.[4]

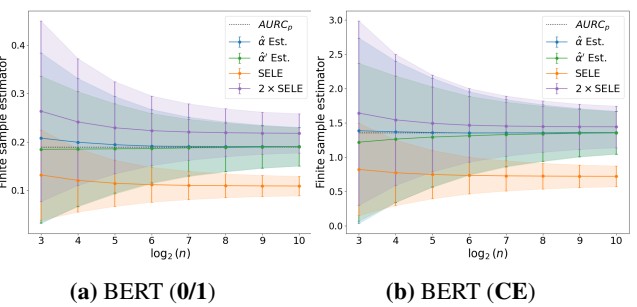

**(a)** BERT (**0/1**)      **(b)** BERT (**CE**)

*Figure 3.* (**Amazon**) Finite sample estimators with **0/1** or **CE** loss. We utilize a pre-trained model and randomly divide the test set into batch samples of size $n$. Subsequently, we compute the mean and std of various estimators applied to these batch samples.

**Measurement of the statistical properties of the estimators.** From Fig. 3, it is observable that with 0/1 loss (accuracy) and increasing sample size, the SELE score tends to underestimate the population $\mathrm{AURC}_p$. Conversely, $2 \times$ SELE tends to overestimate the population $\mathrm{AURC}_p$. The plug-in estimator with $\hat{\alpha}'$ empirically serves as a lower bound for that with $\hat{\alpha}$, supporting the correctness of our theoretical results. As the sample size grows, both estimators progressively converge to the population $\mathrm{AURC}_p$. Similar trends can also be observed regardless of the 0/1 loss or CE loss in Fig. S15-S19. Furthermore, a comparison between Fig. 3(a) and 3(b) reveals that using CE loss rather than 0/1 loss results in a different magnitude of variance and bias in the estimators. The bias plots in Figures S9-S12 show similar findings. In Fig.4, the MAE of the plug-in estimators con-

---

[4]Code is available at https://github.com/han678/AsymptoticAURC.

*Table 1.* Summary of population $AURC_p$ (expressed as mean $\pm$ standard deviation, scaled by $10^{-2}$) of the test set for models fine-tuned with various loss functions. The $AURC_p$ is calculated for each model architecture based on the fine-tuned results aggregated from five different seeds, each using the same pre-trained model.

| Model | CIFAR10 | | | | CIFAR100 | | | |
|---|---|---|---|---|---|---|---|---|
| | CE | SELE | $\hat{\alpha}$ Est. | $\hat{\alpha}'$ Est. | CE | SELE | $\hat{\alpha}$ Est. | $\hat{\alpha}'$ Est. |
| ResNet18 | $4.967_{\pm 0.038}$ | $\mathbf{4.470_{\pm 0.030}}$ | $4.473_{\pm 0.030}$ | $4.471_{\pm 0.030}$ | $6.648_{\pm 0.021}$ | $6.577_{\pm 0.011}$ | $\mathbf{6.532_{\pm 0.012}}$ | $6.533_{\pm 0.014}$ |
| ResNet34 | $6.464_{\pm 0.036}$ | $\mathbf{5.661_{\pm 0.039}}$ | $5.652_{\pm 0.036}$ | $5.651_{\pm 0.036}$ | $6.023_{\pm 0.016}$ | $5.862_{\pm 0.012}$ | $\mathbf{5.825_{\pm 0.011}}$ | $5.826_{\pm 0.011}$ |
| ResNet50 | $8.318_{\pm 0.002}$ | $\mathbf{7.892_{\pm 0.046}}$ | $7.921_{\pm 0.047}$ | $7.918_{\pm 0.049}$ | $6.225_{\pm 0.009}$ | $6.043_{\pm 0.015}$ | $\mathbf{6.007_{\pm 0.008}}$ | $6.007_{\pm 0.009}$ |
| VGG16BN | $7.922_{\pm 0.002}$ | $\mathbf{7.010_{\pm 0.018}}$ | $7.064_{\pm 0.014}$ | $7.060_{\pm 0.015}$ | $10.790_{\pm 0.001}$ | $10.586_{\pm 0.029}$ | $\mathbf{10.559_{\pm 0.029}}$ | $10.560_{\pm 0.030}$ |
| VGG19BN | $9.813_{\pm 0.192}$ | $\mathbf{8.475_{\pm 0.061}}$ | $8.528_{\pm 0.059}$ | $8.524_{\pm 0.059}$ | $10.633_{\pm 0.001}$ | $10.421_{\pm 0.026}$ | $10.393_{\pm 0.025}$ | $\mathbf{10.391_{\pm 0.024}}$ |
| WideResNet28x10 | $4.137_{\pm 0.046}$ | $3.867_{\pm 0.049}$ | $3.864_{\pm 0.049}$ | $\mathbf{3.863_{\pm 0.049}}$ | $5.912_{\pm 0.652}$ | $\mathbf{5.607_{\pm 0.707}}$ | $5.836_{\pm 0.652}$ | $5.607_{\pm 0.707}$ |

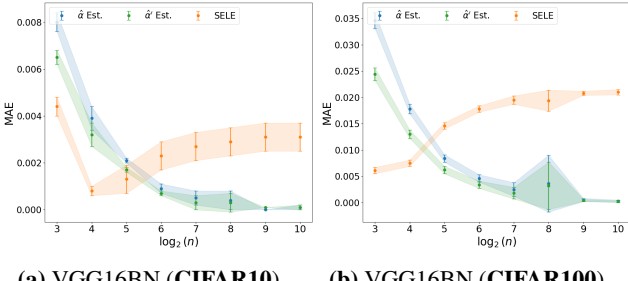

**(a)** VGG16BN (**CIFAR10**)  **(b)** VGG16BN (**CIFAR100**)

*Figure 4.* (**CIFAR10/100**) MAE of different finite sample estimators evaluated with **0/1** loss. For each model architecture, we compute the mean and std of the MAE across five distinct pre-trained models. The MAE for each model is calculated using batch samples divided from the test set. More results can be found in Figs. S13-S14.

sistently decreases as the sample size increases. However, the SELE score does not always exhibit this trend. Its performance lacks stability compared to the plug-in estimators and can even be worse as sample size increases. Results shown in Fig. 5 also indicate a declining trend in the MSE of the plug-in estimators on the ImageNet dataset as the sample size increases, regardless of whether 0/1 or CE loss is used. This convergence is not reflected in the SELE scores, as expected. Similar MSE results are also observed across other model architectures for the CIFAR10/100 and Amazon datasets (see Fig. S20-S25). Although $\hat{\alpha}$ theoretically exhibits lower MSE in weight estimation compared to $\hat{\alpha}'$, its corresponding plug-in estimator empirically achieves an even higher MSE than that of $\hat{\alpha}$.

**Influence of the CSFs.** We also examine the impact of various CSFs on the estimators to provide a thorough evaluation of the metrics. Specifically, we consider MSP, Negative Entropy, MaxLogit, Softmax Margin, MaxLogit-$\ell_2$ norm, and Negative Gini Score, as outlined in Table S2. The sample size is set to 128. We report the results for Amazon and ImageNet datasets in Fig. S26-S28. As indicated in Fig. 6, both plug-in estimators exhibits lower bias compared to other estimators across various CSFs. The 2×SELE score is more likely to overestimate $AURC_p$, but this is not always

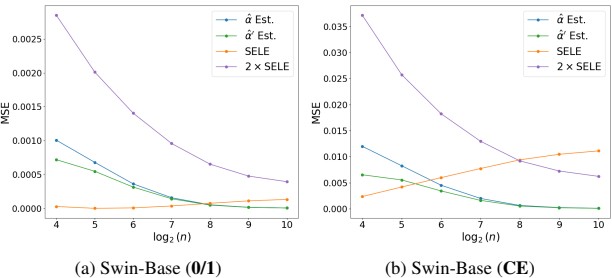

**(a)** Swin-Base (**0/1**)    **(b)** Swin-Base (**CE**)

*Figure 5.* (**ImageNet**) MSE of finite sample estimators with **0/1** or **CE** loss. For each model architecture, we calculate the MSE of the estimators using a pre-trained model on batch samples derived from the test set.

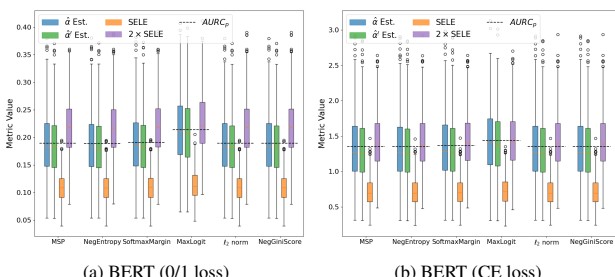

**(a)** BERT (0/1 loss)    **(b)** BERT (CE loss)

*Figure 6.* (**Amazon**) Finite sample estimators with different CSFs.

the case, as shown in Fig. 6(b). The SELE score is substantially lower than the population $AURC_p$ across various CSFs in our evaluations. We can also observe that compared to CSFs, these finite sample estimators are more sensitive to the choice of loss functions. When using 0/1 loss, they display lower variance than CE loss.

**Training a selective classifier.**

We can finetune our pre-trained model using these finite sample estimators as a loss function. The MSP is employed as the CSF when applying the metrics. The models in Table 1 are fine-tuned on the training set using these estimators incorporated with CE loss over 30 epochs, using a learning rate of $10^{-3}$. We set the training batch size to be 128. Additionally, we present the results for both CE loss and SELE score, as detailed in Table 1 for the CIFAR10/100

dataset. As indicated by Table 1, training with these estimators can effectively optimize the $AURC_p$ compared with the CE. Moreover, training with SELE loss also accelerates the optimization of $AURC_p$ compared with CE optimization.

## 5. Conclusion and Future Work

In this work, we revisit the definition of empirical AURC and propose the population AURC from a statistical perspective, along with an equivalent expression that can be interpreted as a reweighted risk function. Subsequently, we introduce a plug-in estimator for population AURC, characterized by a biased weight estimator. Additionally, we provide an alternative derivation of this and another plug-in estimator using the Monte Carlo method. We rigorously analyze the statistical properties of these Monte Carlo-derived weight estimators, including their bias, MSE, and consistency, and establish their convergence rate to be $\mathcal{O}(\sqrt{\ln(n)/n})$. To validate our theoretical results, we evaluate the estimator across various state-of-the-art neural network models and widely-used datasets. Both plug-in estimators exhibit better performance compared to the SELE score. Finally, we have demonstrated that the combination of good statistical convergence and efficient computation make them suitable training objectives for directly fine-tuning networks to minimize AURC.

In this paper, our primary focus is on the estimation of AURC for a fixed model. For completeness, we discuss in the appendix the scenario in which a Bayesian model is considered, specifically when $f \sim \mathbb{P}(f|\mathcal{D})$. We anticipate that these directions will inspire further research. Additionally, investigating the performance of estimators under distribution shift or in the context of imbalanced datasets represents a promising avenue for future work. We also encourage studies that adapt estimators of the form developed in this paper to these settings.

## Impact Statement

This paper presents work whose goal is to advance the field of Machine Learning. There are many potential societal consequences of our work, none which we feel must be specifically highlighted here.

## Acknowledgement

This research received funding from the Flemish Government (AI Research Program) and the Research Foundation - Flanders (FWO) through project number G0G2921N. HZ is supported by the China Scholarship Council. We acknowledge EuroHPC JU for awarding the project ID EHPC-BEN-2024B10-050 and EHPC-BEN-2025B22-037 access to the EuroHPC supercomputer LEONARDO, hosted by CINECA (Italy) and the LEONARDO consortium.

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

## A. Appendix

### A.1. Additional Proofs

**Proposition A.1** (Consistency of $\hat{\alpha}_i$). *Assume $\beta_i$ is the population rank percentile of the observation $(x_i, y_i)$ ranked by CSF. Under this definition, the parameter $\hat{\alpha}_i$ is consistent, converging to the limit*

$$\lim_{n \to \infty} \hat{\alpha}_i = -\ln(1 - \beta_i).$$

*Proof.* Given the sample size $n$ and sample rank $r_i$, let us set $r_i = \beta_i' n$ for $\beta_i' \in (0, 1)$ and take the limit

$$
\begin{aligned}
\lim_{n \to \infty} [\mathrm{H}_n - \mathrm{H}_{n-\beta_i n}] &= \lim_{n \to \infty} [\psi(n + 1) - \psi(n - \beta_i' n + 1)] \\
&= \lim_{n \to \infty} \left[ \ln(n + 1) - \frac{1}{2(n + 1)} - \ln(n - \beta_i' n + 1) + \frac{1}{2(n - \beta_i' n + 1)} \right] \\
&= \lim_{n \to \infty} \left[ -\frac{1}{2(n + 1)} + \frac{1}{2(n - \beta_i' n + 1)} - \ln(1 - \beta_i') + \frac{\beta_i'}{n + 1}) \right] \\
&= -\lim_{n \to \infty} \ln(1 - \beta_i') \\
&= -\ln(1 - \beta_i)
\end{aligned}
\tag{27}
$$

where the 2nd equation was obtained using the asymptotic result that $\psi(n) \to \ln n - \frac{1}{2n}$ as $n \to \infty$. $\square$

**Proposition A.2** (**Bias of $\hat{\alpha}_i^{se}$**). *The bias of the the weight estimator $\hat{\alpha}_i^{se}$ corresponding to the SELE score is given by*

$$\mathrm{Bias}\,(\hat{\alpha}_i^{se}|G(x_i) = \beta_i) = \sum_{i=1}^{n} \frac{i}{n} C_{n-1}^{i-1} \beta_i^{i-1} (1 - \beta_i)^{n-i} + \ln(1 - \beta_i).$$

*Proof.* From Sec. A.2, the expected $\hat{\alpha}_i^{se}$ associated with $(x_i, y_i)$ is given by

$$\mathbb{E}\,[\hat{\alpha}_i^{se}|G(x_i) = \beta_i] = \sum_{i=1}^{n} \frac{i}{n} \mathrm{Pr}(r_i = i|G(x_i) = \beta_i).$$

Since $\mathrm{Pr}(r_i|G(x_i) = \beta_i)$ is a binomial distribution $\mathrm{Bin}(n - 1, \beta_i)$, we obtain:

$$\mathrm{Bias}\,(\hat{\alpha}_i'|G(x_i) = \beta_i) = \mathbb{E}\,[\hat{\alpha}_i'|G(x_i) = \beta_i] - \alpha_i = \sum_{i=1}^{n} \frac{i}{n} C_{n-1}^{i-1} \beta_i^{i-1} (1 - \beta_i)^{n-i} + \ln(1 - \beta_i) \tag{28}$$

which conclude our proof. $\square$

**Proposition A.3** (MSE of $\hat{\alpha}_i'$). *The MSE of $\hat{\alpha}_i'$ is given by*

$$\mathrm{MSE}(\hat{\alpha}_i') = \psi'(n + 1 - r_i) - \psi'(n + 1) + \left( \ln\left(1 - \frac{r_i}{n + 1}\right) + \mathrm{H}_n - \mathrm{H}_{n-r_i} \right)^2 \asymp \frac{\beta_i}{n(1 - \beta_i) + 1}. \tag{29}$$

*Proof.* From the result in Proposition 3.1, we calculate the MSE as follows:

$$\mathrm{MSE}(\hat{\alpha}_i') = \mathbb{E}_{\beta_i} \left[ (\hat{\alpha}_i' + \ln(1 - \beta_i))^2 \right],$$

which becomes:

$$\mathrm{MSE}(\hat{\alpha}_i') = \int_0^1 \left( -\ln\left(1 - \frac{r_i}{n + 1}\right) + \ln(1 - \beta_i) \right)^2 d\mathrm{P}(\beta_i).$$

We can break this into two parts, denoted $M$ and $N$:

$$\mathrm{MSE}(\hat{\alpha}_i') = M + N,$$

where

$$M = \int_0^1 \ln^2(1 - \beta_i) \, d\mathrm{P}(\beta_i) = (\mathrm{H}_n - \mathrm{H}_{n-r_i})^2 + \psi'(n+1-r_i) - \psi'(n+1),$$

and

$$N = \ln^2\left(1 - \frac{r_i}{n+1}\right) + 2\ln\left(1 - \frac{r_i}{n+1}\right)(\mathrm{H}_n - \mathrm{H}_{n-r_i}).$$

Here, $d\mathrm{P}(\beta_i)$ refers to the integration with respect to the probability measure induced by $\beta_i \sim \mathrm{Beta}(r_i, n+1-r_i)$. Now, by combining $M$ and $N$, we obtain the MSE:

$$\mathrm{MSE}(\hat{\alpha}_i') = \psi'(n+1-r_i) - \psi'(n+1) + Q^2,$$

where

$$Q = \ln\left(1 - \frac{r_i}{n+1}\right) + \mathrm{H}_n - \mathrm{H}_{n-r_i}.$$

Using the inequalities of the harmonic numbers, $\gamma + \ln(n) \le \mathrm{H}_n \le \ln(n+1) + \gamma$, we obtain the following bounds:

$$0 \le \mathrm{H}_n - \mathrm{H}_{n-r_i} \le \ln(n+1) - \ln(n-r_i).$$

This leads to the upper bound for $Q$:

$$Q \le \ln\left(1 + \frac{1}{n-r_i}\right) \le \frac{1}{n-r_i},$$

since it holds that $\ln(1+x) \le x$ for $x > -1$. For the remaining term, we use the same approach as in Proposition 3.5 and obtain the following estimate:

$$\psi'(n+1-r_i) - \psi'(n+1) \asymp \mathcal{O}\left(\frac{\beta_i}{n(1-\beta_i)+1}\right).$$

Combining this with the upper bound for $Q$, we obtain the final bound for the MSE:

$$\mathrm{MSE}(\hat{\alpha}_i') \asymp \frac{\beta_i}{n(1-\beta_i)+1} \quad \forall \beta_i \in (0,1) \tag{30}$$

as the MSE is dominated by this remaining term. $\qquad \square$

**Proposition A.4** ($\frac{1}{n}\sum_{i=1}^n \mathrm{MSE}(\hat{\alpha}_i)$ or $\frac{1}{n}\sum_{i=1}^n \mathrm{MSE}(\hat{\alpha}_i')$)**.** *The average of the sum of the MSEs of the weight estimators* $\hat{\alpha}_i$ *or* $\hat{\alpha}_i'$ *is tightly bounded by* $\mathcal{O}\left(\frac{\ln(n)}{n}\right)$.

*Proof.* From result in Proposition 3.5, we derive:

$$\begin{aligned}
\frac{1}{n}\sum_{i=1}^n \mathrm{MSE}(\hat{\alpha}_i) &\to \frac{1}{n}\sum_{i=1}^n \frac{\beta_i}{n(1-\beta_i)+1} \\
&\to \int_0^1 \frac{\beta}{n(1-\beta)+1} d\beta \\
&= \frac{(n+1)\ln(n+1)}{n^2} - \frac{1}{n} \\
&= \mathcal{O}\left(\frac{\ln(n)}{n}\right)
\end{aligned} \tag{31}$$

We could obtain the same results for $\hat{\alpha}_i'$, thereby concluding our proof. $\qquad \square$

## A.2. SELE score

Franc et al. (2023) gave another expression of the empirical AURC in Eq. (10) that can be interpreted as an arithmetic mean of the empirical selective risks corresponding to the coverage spread evenly over the interval $[0, 1]$ with step $\frac{1}{n}$. In addition, they proposed a coarse lower bound, referred to as the SELE score given by

$$\Delta_{\text{sele}}(f) = \frac{1}{n^2} \sum_{i=1}^{n} \sum_{j=1}^{n} \ell\left(f\left(x_i\right), y_i\right) \mathbb{I}\left[g\left(x_i\right) \geq g(x_j)\right], \tag{32}$$

as an alternative to the empirical AURC. They claim that $2\Delta_{\text{sele}}(f)$ is an upper bound to the empirical AURC, but in the Appendix A.2, we demonstrate that this is not the case. This naive implementation of this metric requires $O(n^2)$ operations but we can rewrite it in a form that can be computed in $\mathcal{O}(n \ln(n))$ using the trick for empirical AURC:

$$\Delta_{\text{sele}}(f) = \sum_{i=1}^{n} \frac{r_i}{n^2} \ell\left(f\left(x_i\right), y_i\right). \tag{33}$$

For this metric, $\hat{\alpha}_i^{se} = \frac{r_i}{n}$ serves as the estimate for $\alpha(\cdot)$. For the SELE score in Eq. (33), we can show that $2\Delta_{\text{sele}}(f)$ does not always serve as an upper bound for empirical AURC, meaning that (Franc et al., 2023, Theorem 8) does not hold. We could find a simple counterexample s.t. the following inequality holds

$$\widehat{\text{AURC}}_p(f) > 2\Delta_{\text{sele}}(f). \tag{34}$$

Given a dataset of 5 observations sorted according to the CSF $\{x_i, y_i\}_{i=1}^{5}$, it is possible to find a classifier $f$ s.t. $\ell\left(f\left(x_i\right), y_i\right) = 0$ for $i = 1 \cdots 4$ and $\ell\left(f\left(x_5\right), y_5\right) > 0$. Then we would have:

$$\hat{\alpha}_5 = H_5 - H_0 \approx 2.2833 \geq 2\hat{\alpha}_5^{se} = 2, \tag{35}$$

which leads to

$$\widehat{\text{AURC}}_p(f) = \sum_{i=1}^{5} \hat{\alpha}_i \ell\left(f\left(x_i\right), y_i\right) \geq \sum_{i=1}^{5} 2\hat{\alpha}_i^{se} \ell\left(f\left(x_i\right), y_i\right) = 2\Delta_{\text{sele}}(f). \tag{36}$$

In their proof of Theorem 8, since $\frac{a_i}{b_i}$ is a non-decreasing sequence so they cannot use Lemma 15 to derive their results.

## A.3. A Generalized Approach to Epistemic Risk for Bayesian Models

In this work, we mainly focus on the estimation of AURC for a fixed model. However, when considering a Bayesian model with $f \sim \mathbb{P}(f|\mathcal{D})$, a natural approach is to consider the following expectation:

$$\mathbb{E}_{f \sim \mathbb{P}(f|\mathcal{D})}[\text{AURC}(f)] = \mathbb{E}_{f \sim \mathbb{P}(f|\mathcal{D})}\left[\mathbb{E}_x\left[\mathcal{R}\left(f, g\right) | \tau = g(x)\right]\right] \tag{37}$$

where $\tau$ is the threshold value. This can be computed directly using our method with Monte Carlo sampling. By applying Fubini's Theorem, we can exchange the order of the two expectations, leading to

$$\mathbb{E}_{f \sim \mathbb{P}(f|\mathcal{D})}[\text{AURC}(f)] = \mathbb{E}_x\left[\mathbb{E}_{f \sim \mathbb{P}(f|\mathcal{D})}\left[\mathcal{R}\left(f, g\right) | \tau = g(x)\right]\right]. \tag{38}$$

Since $g(x)$ depends on $f(x)$ under the posterior distribution $\mathbb{P}(f|\mathcal{D})$, and the predictions are made through a full Bayesian framework, this formulation allows the evaluation of AURC in a way analogous to the standard AURC for a fixed model. We can envision several ways to define potential quantities of interest based on model uncertainty. Many of these quantities could be potentially connected to AURC and epistemic risk, and exploring these relationships could open up valuable avenues for further investigation.

## A.4. Confidence Score Functions (CSFs)

The CSFs are generally defined as functions of the predicted probabilities $\mathbf{p}_i$, which are the outputs by passing the logits $\mathbf{z}$ produced by the model for the input $x$ through the softmax function $\sigma(\cdot)$, expressed as $\sigma(\mathbf{z}) \in \mathbb{R}^K$. The specific forms of these CSFs are outlined as follows:

*Table S2.* Commonly Used CSFs

| Method | Equation |
|---|---|
| MSP | $g(\mathbf{z}) = \max_{i=1}^{K} \mathbf{p}_i$ |
| MaxLogit | $g(\mathbf{z}) = \max_{i=1}^{K} \mathbf{z}_i$ |
| Softmax Margin | $g(\mathbf{z}) = \mathbf{p}_i - \max_{j \neq i} \mathbf{p}_j$ with $i = \arg\max_{i=1}^{K} \mathbf{p}_i$ |
| Negative Entropy | $g(\mathbf{z}) = \sum_{i=1}^{K} \mathbf{z}_i \log \mathbf{z}_i$ |
| MaxLogit-$\ell_p$ Norm | $g(\mathbf{z}) = \|\mathbf{z}\|_p$ |
| Negative Gini Score | $g(\mathbf{z}) = -1 + \sum_{i=1}^{K} \mathbf{p}_i^2$ |

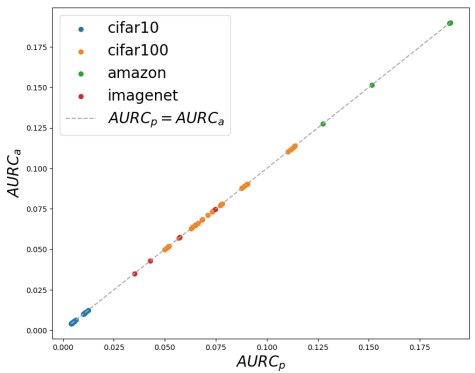

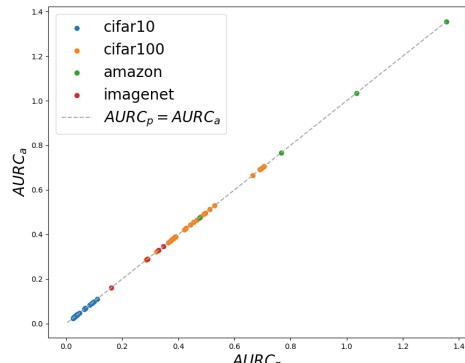

*Figure S6.* (a) 0/1 Loss

*Figure S6.* (b) CE Loss

*Figure S7.* Comparison of AURC$_a$ and AURC$_p$ under different loss functions. The AURC$_p$ and AURC$_a$ are computed across the test set using Eqs. (4) and (8), respectively. Subfigure (a) shows the 0/1 loss, while subfigure (b) depicts the CE loss.

## A.5. Empirical Comparison Between AURC$_a$ and AURC$_p$

In Section 3.4, we demonstrate the theoretical equivalence of these two metrics, and here, we aim to provide an empirical validation of this equivalence. We evaluate the two population AURC metrics using either 0/1 or CE loss across 30 different models on test sets from CIFAR10/100. We also assess these metrics for the previously mentioned models on the test sets from Amazon and ImageNet datasets. The results are reported in Fig. S7, where these two population AURC metrics are shown to be identical to each other. We also assessed the results using a two-sided t-test, which yielded p-values of 0.9981 and 0.998 for the 0/1 and CE loss, respectively. These values suggest that we should accept the null hypothesis that these two metrics are identical.

## A.6. Figures

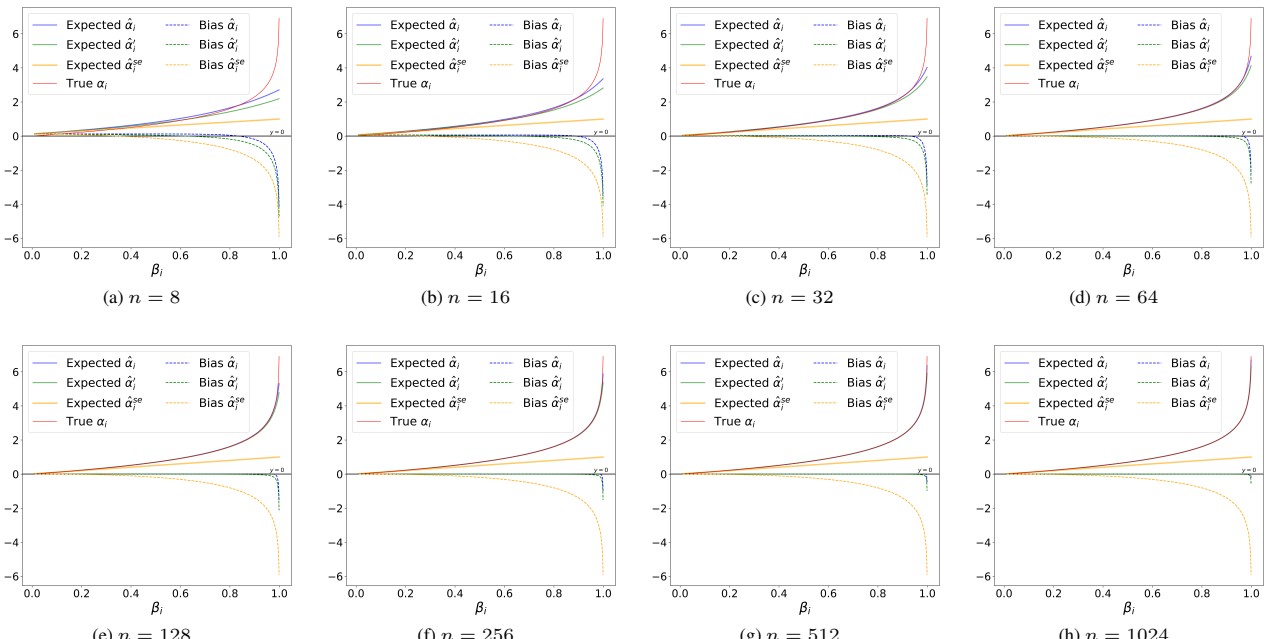

*Figure S8.* The bias of the weights estimator as a function of $\beta$ for different sample size $n$. The bias is computed based on the results in Prop. 3.3, 3.4 and A.2.

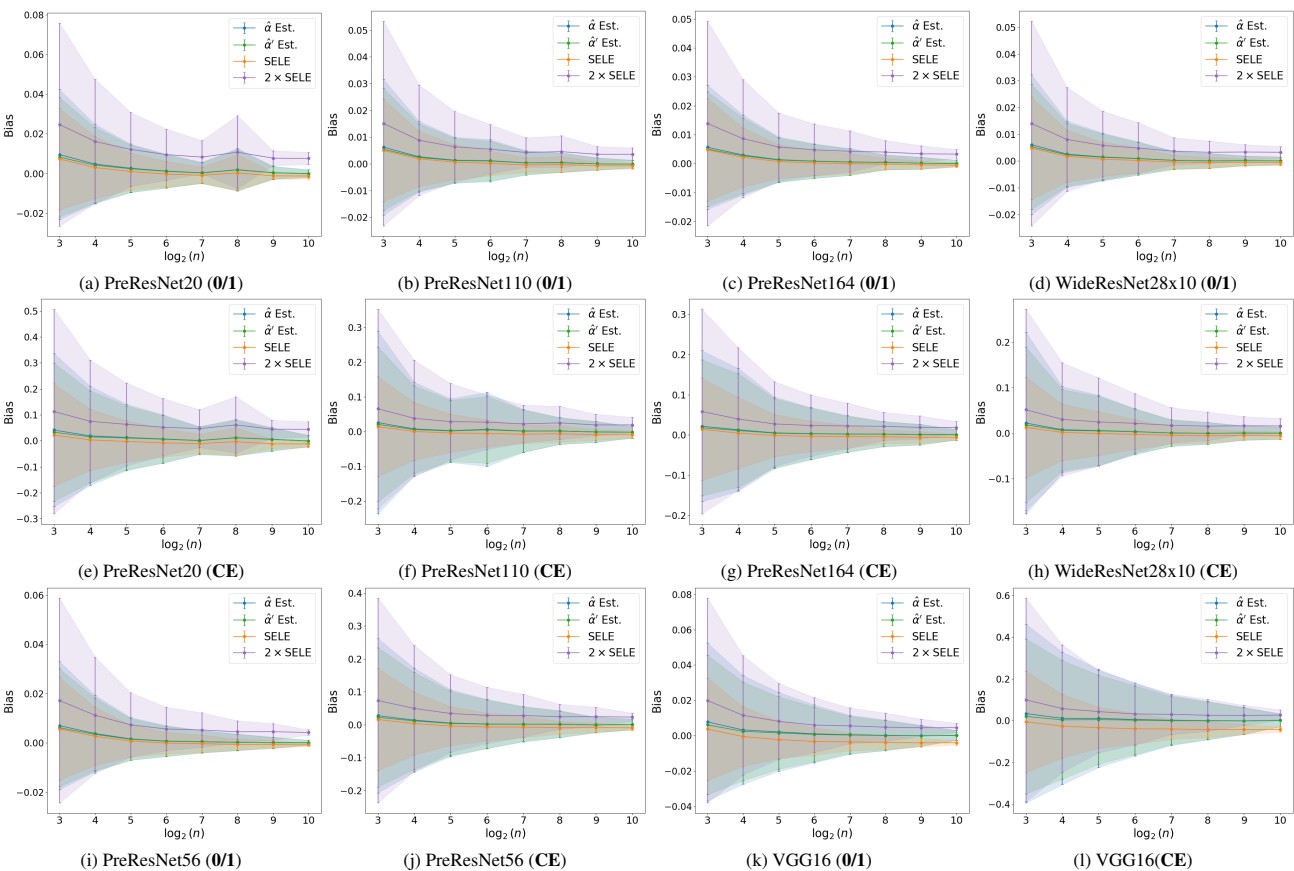

*Figure S9.* (**CIFAR10**) Bias of different finite sample estimators evaluated with **0/1** or **CE** loss. For each model architecture, We utilize a pre-trained model and randomly divide the test set into batch samples of size $n$. Subsequently, we compute the mean and std of bias for various estimators applied to these batch samples.

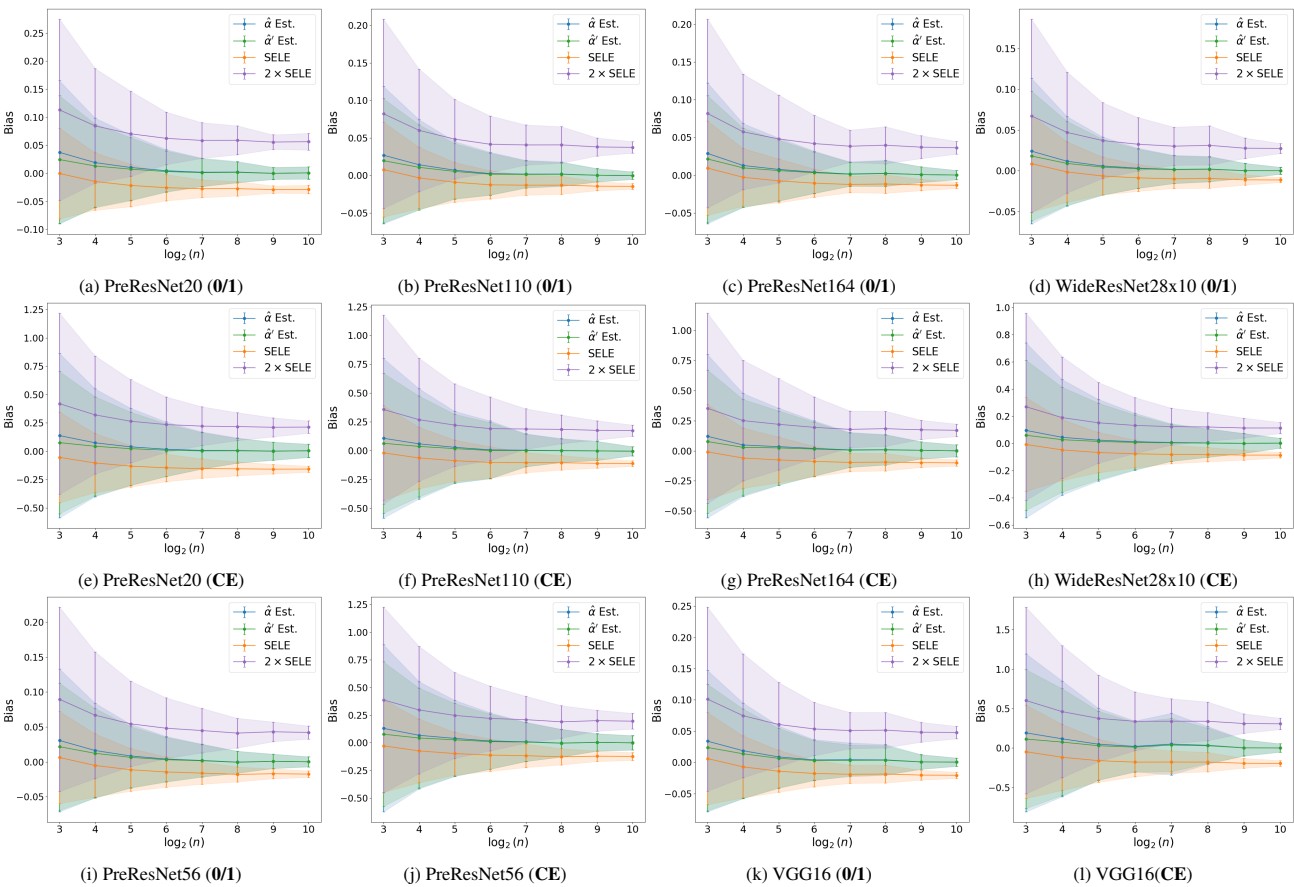

*Figure S10.* (**CIFAR100**) Bias of different finite sample estimators evaluated with **0/1** or **CE** loss. For each model architecture, We utilize a pre-trained model and randomly divide the test set into batch samples of size $n$. Subsequently, we compute the mean and std of bias for various estimators applied to these batch samples.

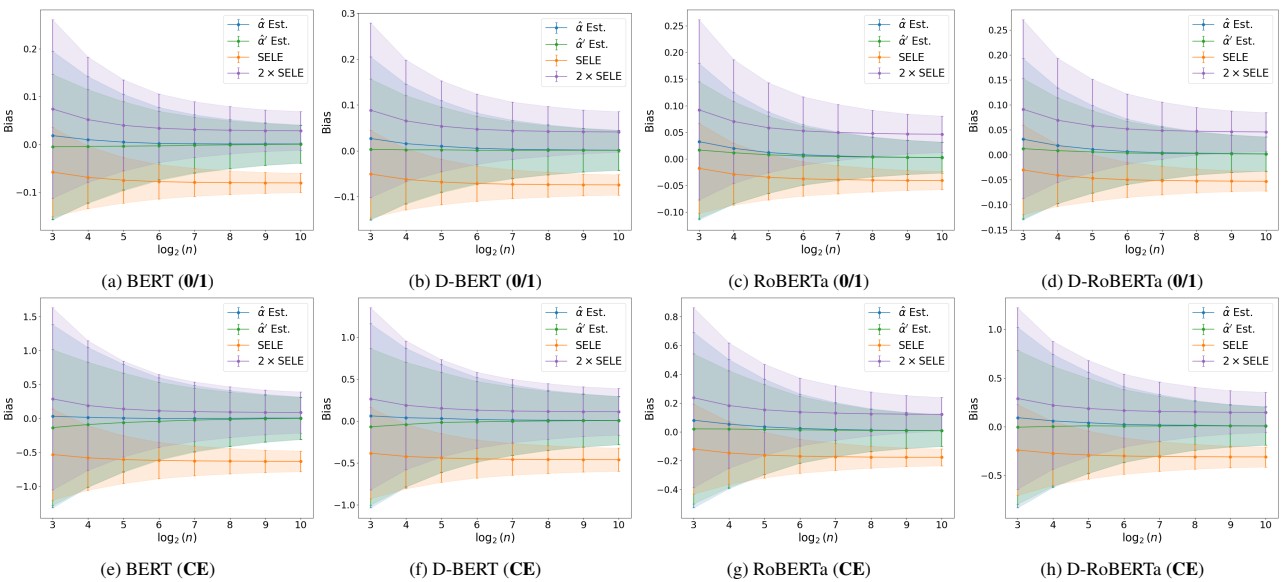

*Figure S11.* (**Amazon**) Bias of different finite sample estimators evaluated with **0/1** or **CE** loss. For each model architecture, We utilize a pre-trained model and randomly divide the test set into batch samples of size $n$. Subsequently, we compute the mean and std of bias for various estimators applied to these batch samples.

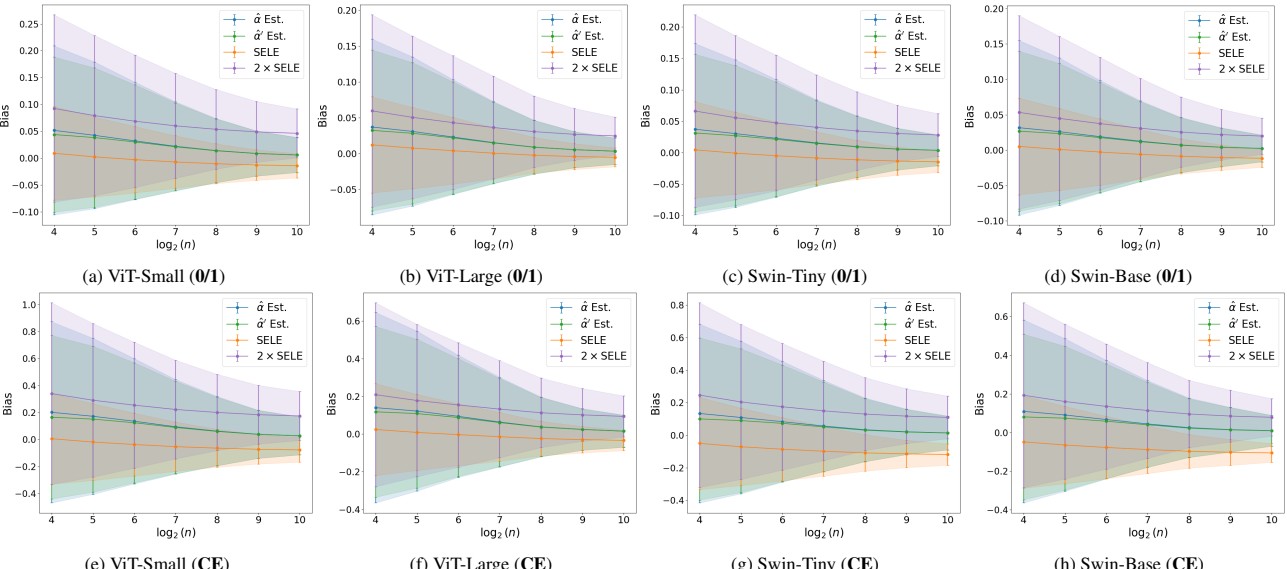

(a) ViT-Small (**0/1**)     (b) ViT-Large (**0/1**)     (c) Swin-Tiny (**0/1**)     (d) Swin-Base (**0/1**)

(e) ViT-Small (**CE**)     (f) ViT-Large (**CE**)     (g) Swin-Tiny (**CE**)     (h) Swin-Base (**CE**)

*Figure S12.* (**ImageNet**) Bias of different finite sample estimators evaluated with **0/1** or **CE** loss. For each model architecture, We utilize a pre-trained model and randomly divide the test set into batch samples of size $n$. Subsequently, we compute the mean and std of bias for various estimators applied to these batch samples.

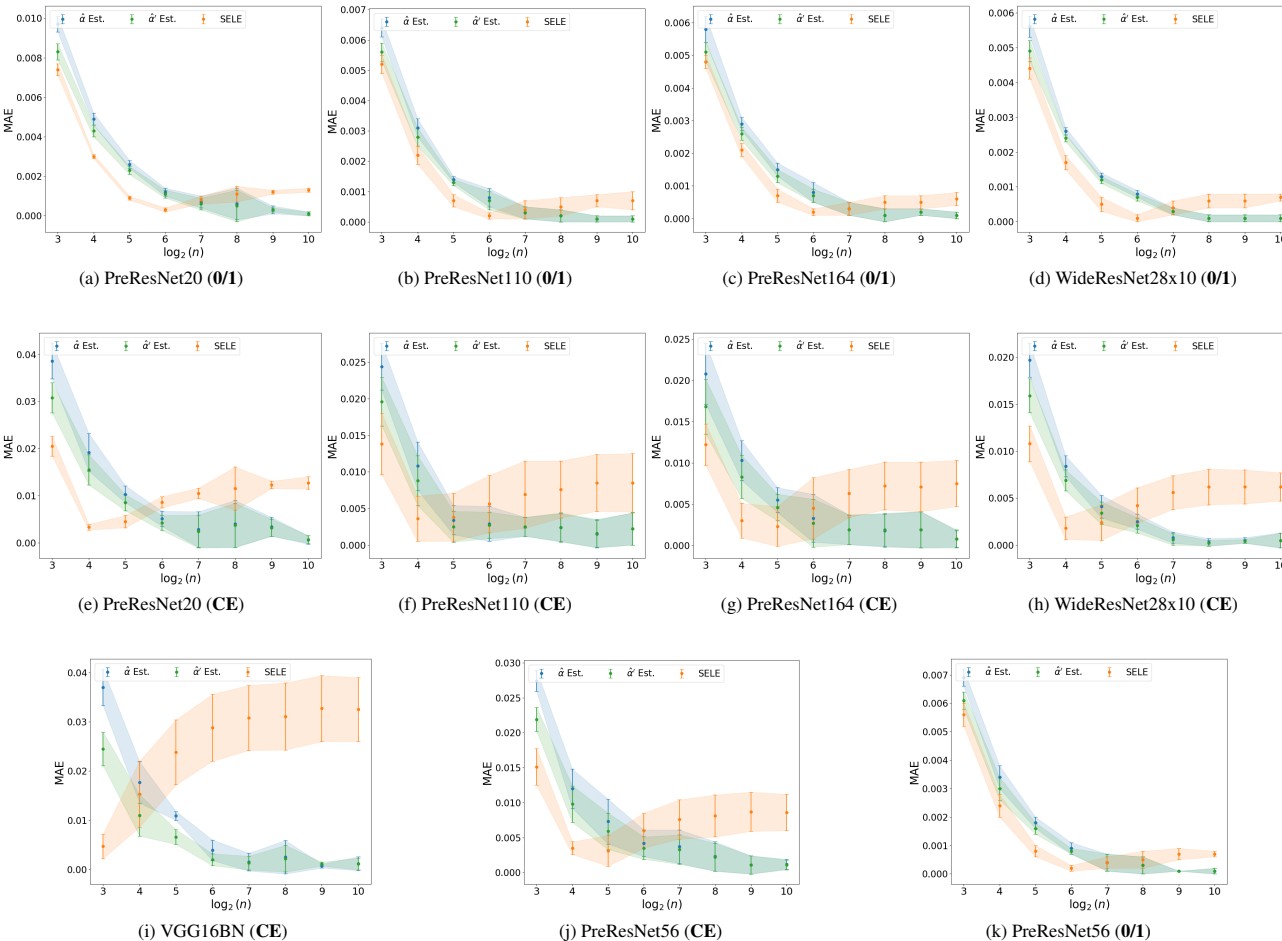

*Figure S13.* (**CIFAR10**) MAE of different finite sample estimators evaluated with **0/1** or **CE** loss. For each model architecture, we compute the mean and std of the MAE across five distinct pre-trained models. The MAE for each model is calculated using batch samples divided from the test set.

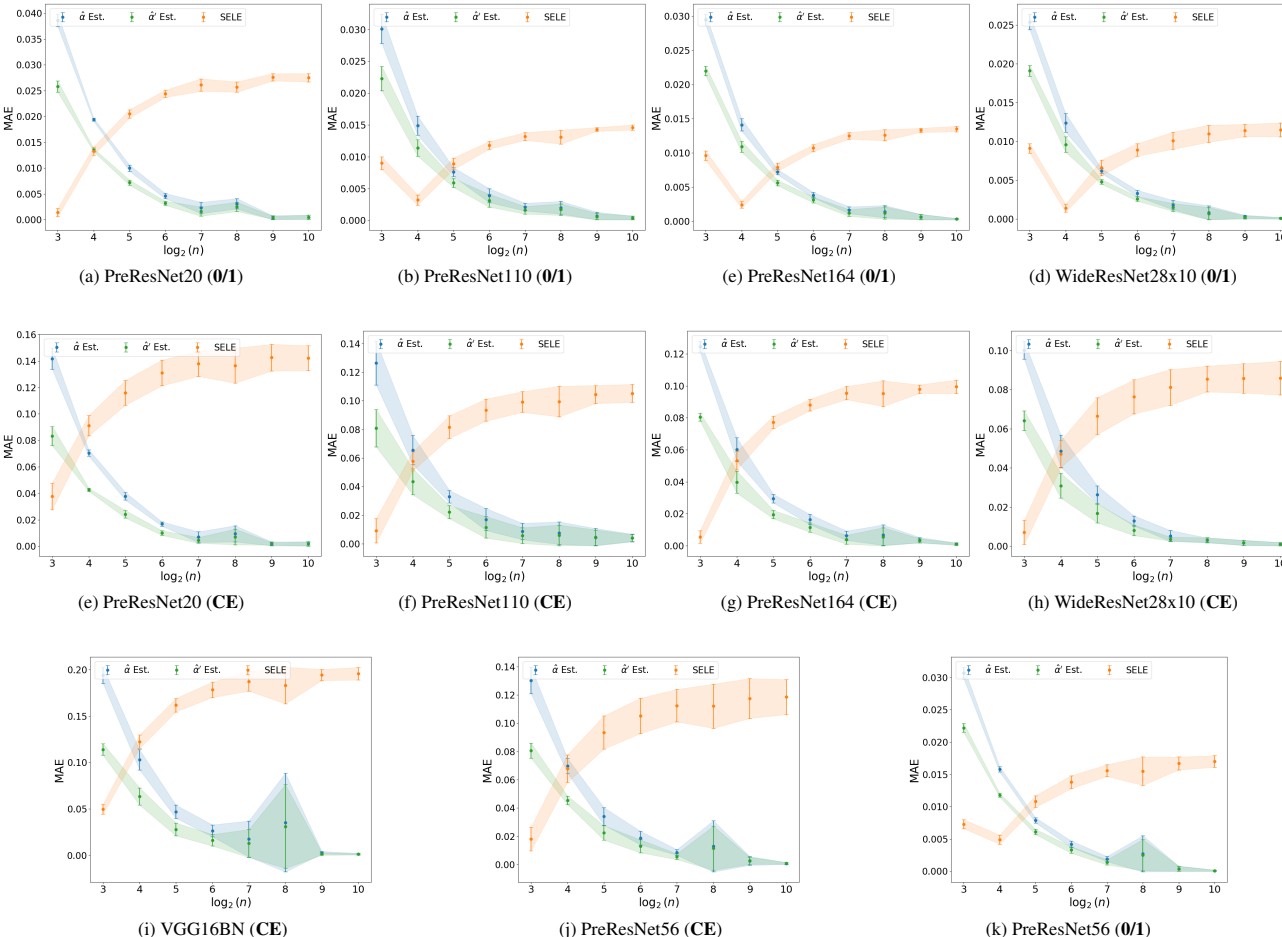

*Figure S14.* (**CIFAR100**) MAE of different finite sample estimators evaluated with **0/1** or **CE** loss. For each model architecture, we compute the mean and std of the MAE across five distinct pre-trained models. The MAE for each model is calculated using batch samples divided from the test set.

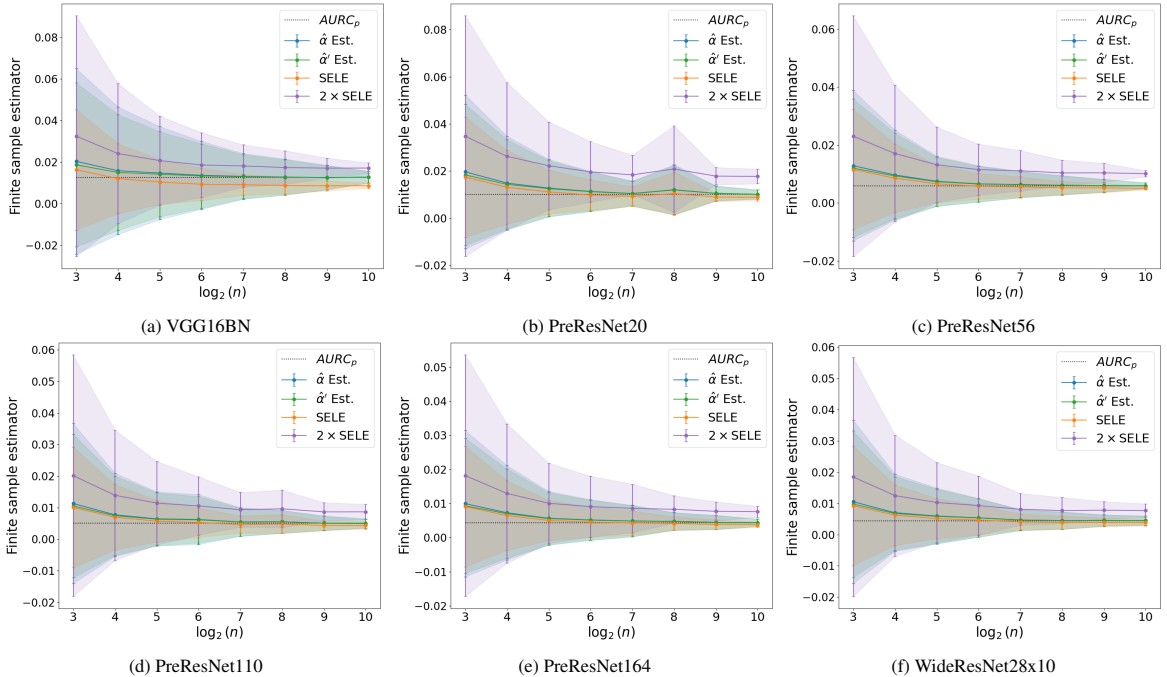

*Figure S15.* Finite sample estimators on **CIFAR10** dataset with **0/1** loss. We utilize a pre-trained model and randomly divide the test set into batch samples of size $n$. Subsequently, we compute the mean and variance of various estimators applied to these batch samples. Additionally, the population $\text{AURC}_p$ is computed across all samples in the test set.

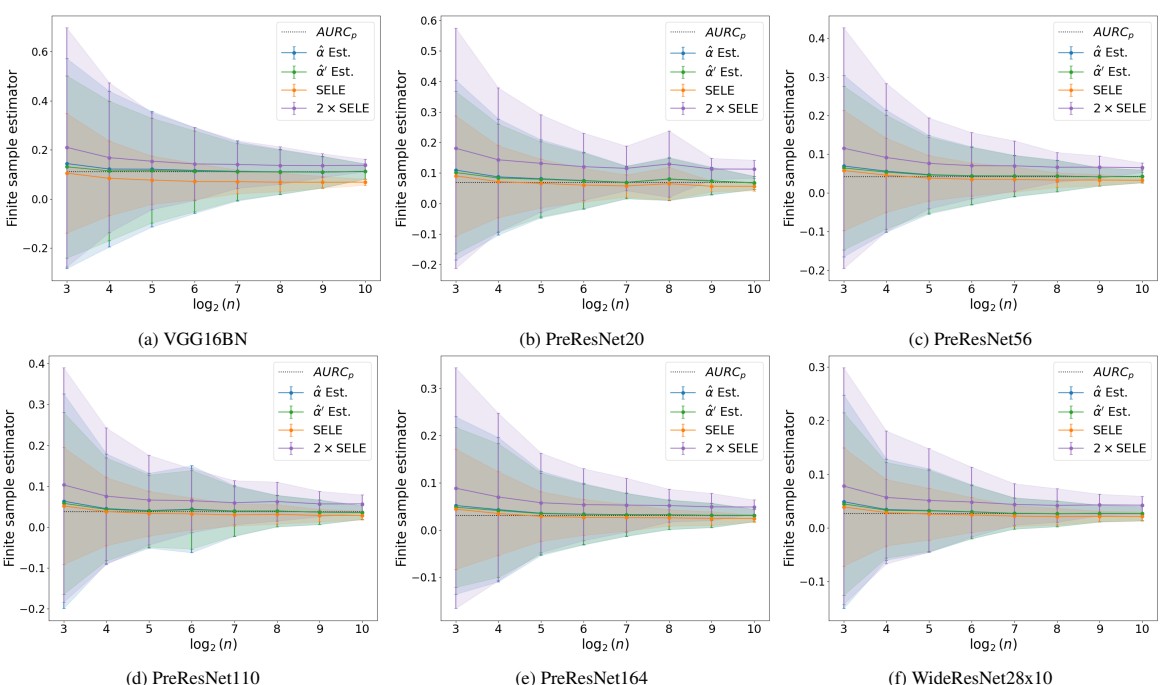

*Figure S16.* Finite sample estimators on **CIFAR10** dataset with **CE** loss. We utilize a pre-trained model and randomly divide the test set into batch samples of size $n$. Subsequently, we compute the mean and variance of various estimators applied to these batch samples. Additionally, the population $\text{AURC}_p$ is computed across all samples in the test set.

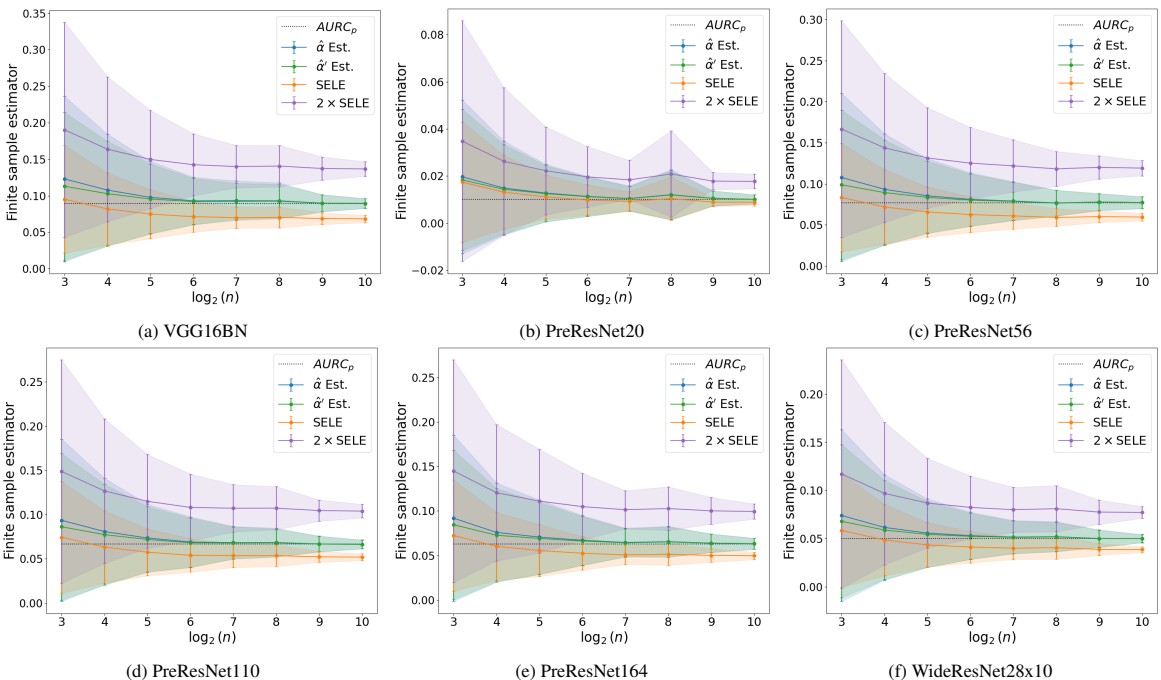

*Figure S17.* Finite sample estimators on the **CIFAR100** dataset with **0/1** loss. We utilize a pre-trained model and randomly divide the test set into batch samples of size $n$. Subsequently, we compute the mean and variance of various estimators applied to these batch samples. Additionally, the population AURC$_p$ is computed across all samples in the test set.

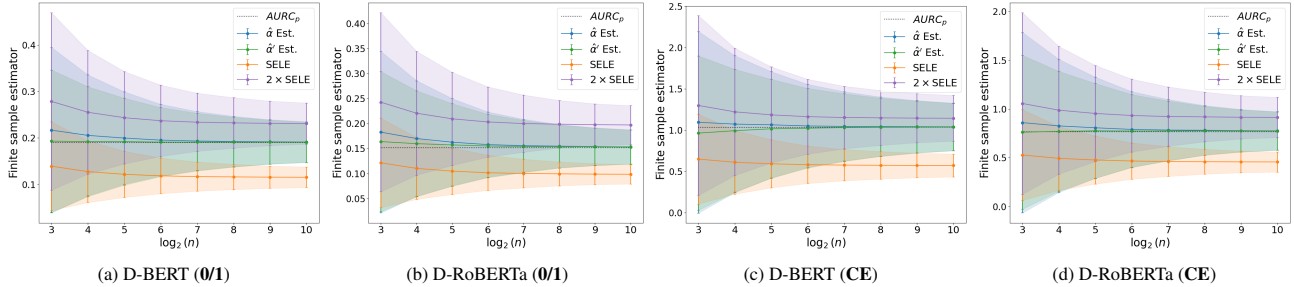

*Figure S18.* (**Amazon**) Finite sample estimators with **0/1** or **CE** loss. We utilize a pre-trained model and randomly divide the test set into batch samples of size $n$. Subsequently, we compute the mean and variance of various estimators applied to these batch samples. Additionally, the population AURC$_p$ is computed across all samples in the test set.

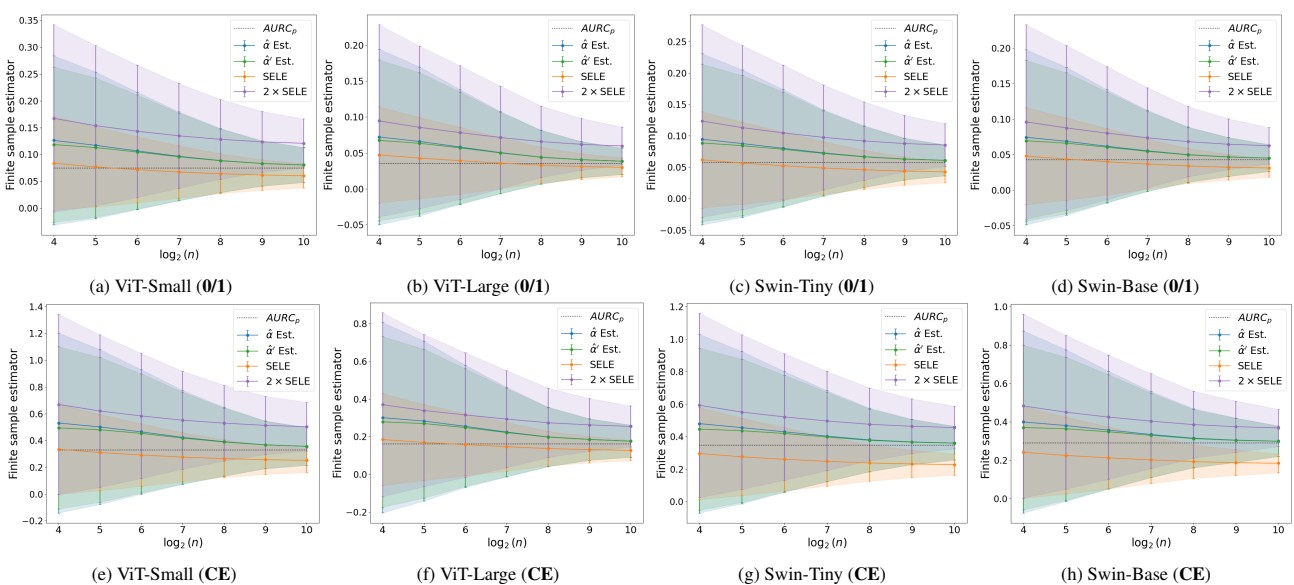

*Figure S19.* (**ImageNet**) Finite sample estimators with **0/1** or **CE** loss. We utilize a pre-trained model and randomly divide the test set into batch samples of size $n$. Subsequently, we compute the mean and variance of various estimators applied to these batch samples. Additionally, the population $\text{AURC}_p$ is computed across all samples in the test set.

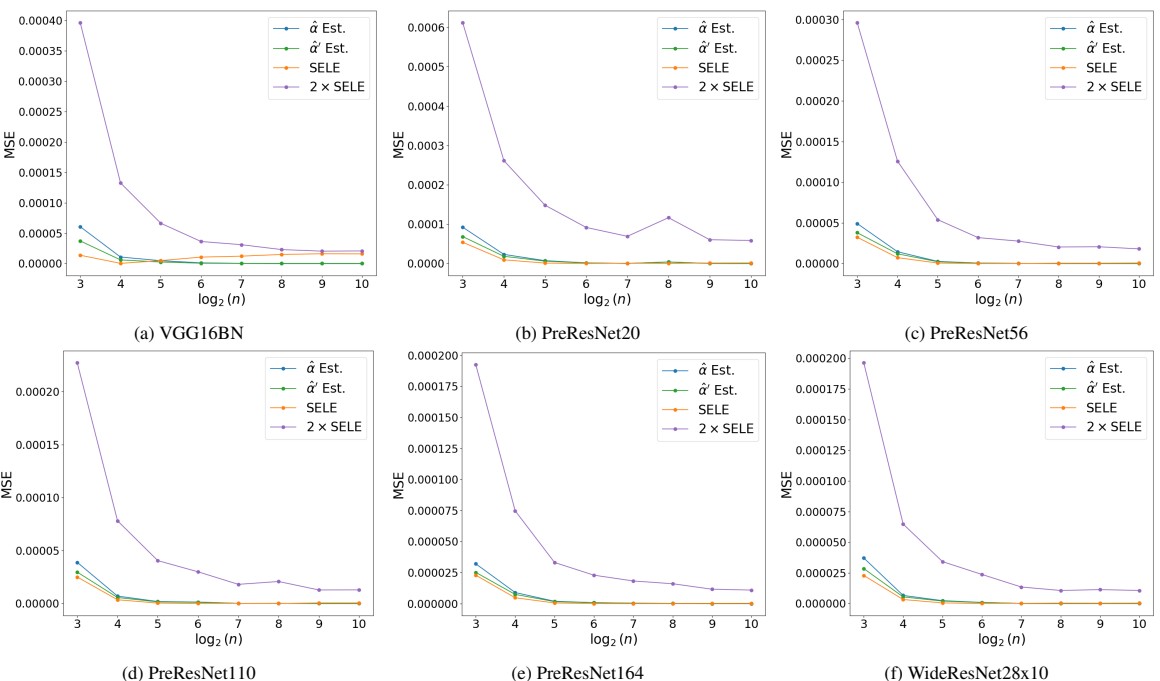

*Figure S20.* (**CIFAR10**) MSE of different finite sample estimators with **0/1** loss. For each model architecture, we calculate the MSE of the estimators using a pre-trained model on batch samples derived from the test set.

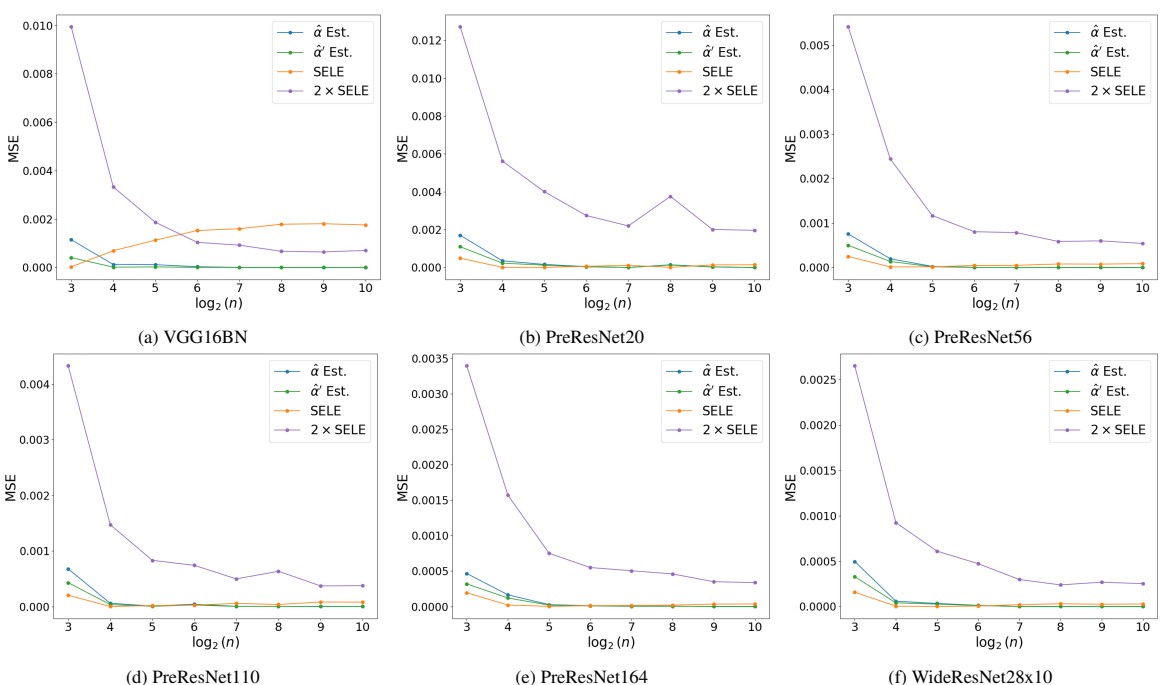

*Figure S21.* (**CIFAR10**) MSE of different finite sample estimators with **CE** loss. For each model architecture, we calculate the MSE of the estimators using a pre-trained model on batch samples derived from the test set.

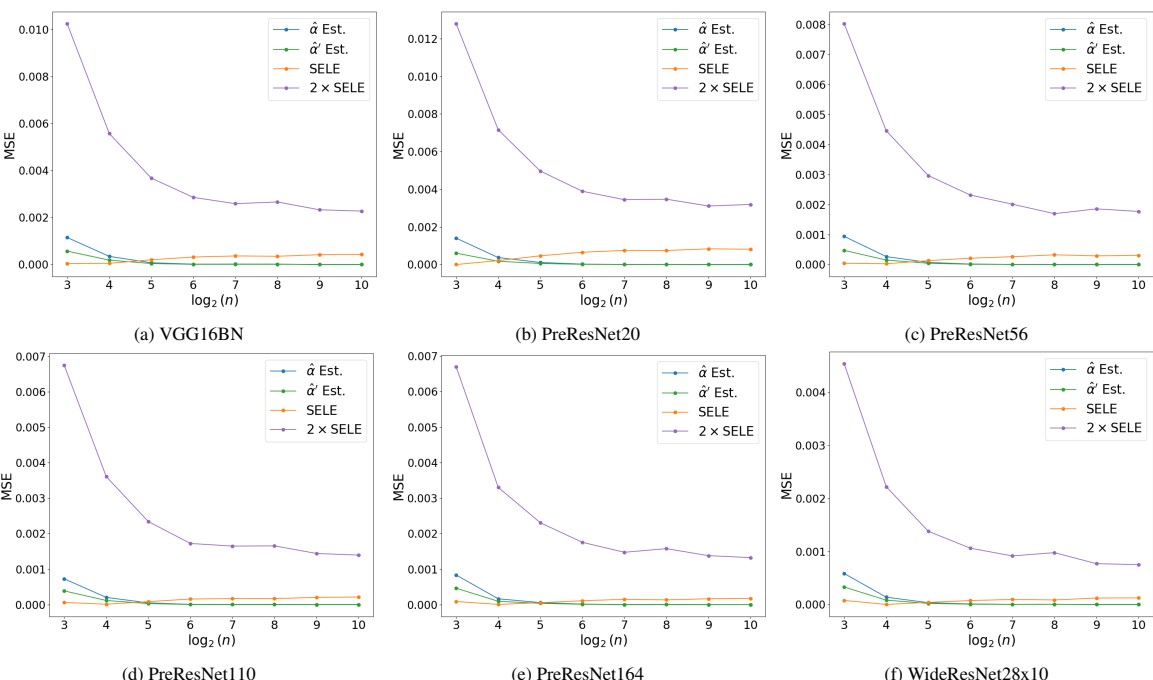

*Figure S22.* (**CIFAR100**) MSE of different finite sample estimators with **0/1** loss. For each model architecture, we calculate the MSE of the estimators using a pre-trained model on batch samples derived from the test set.

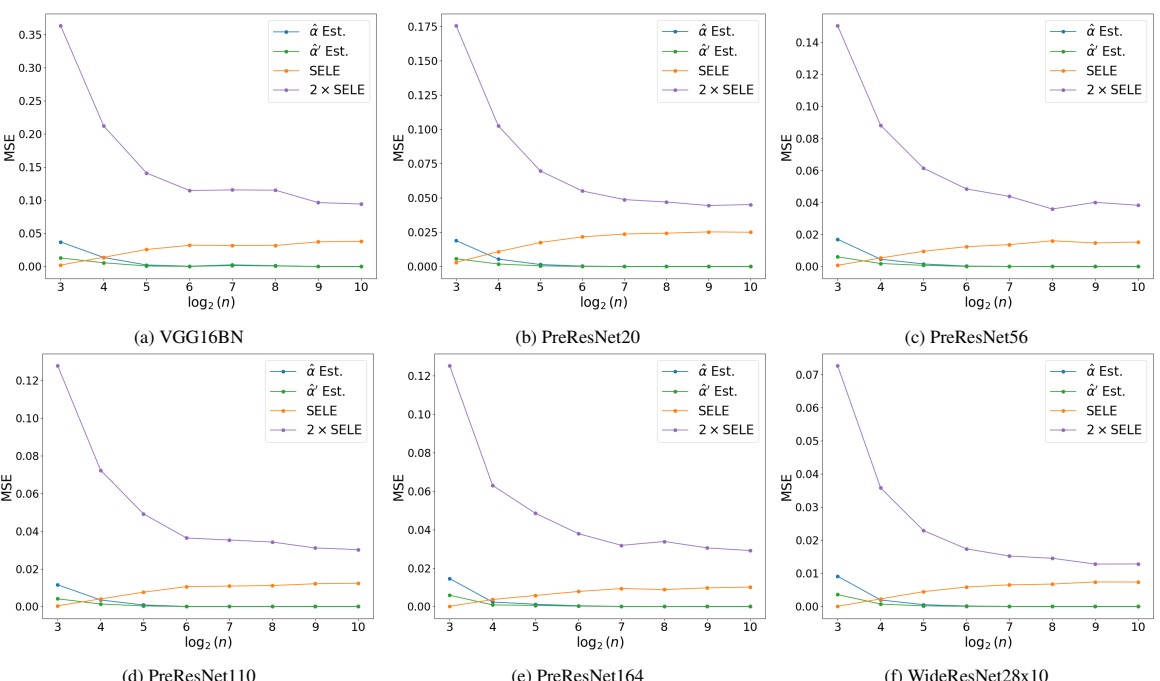

*Figure S23.* (**CIFAR100**) MSE of different finite sample estimators with **CE** loss. For each model architecture, we calculate the MSE of the estimators using a pre-trained model on batch samples derived from the test set.

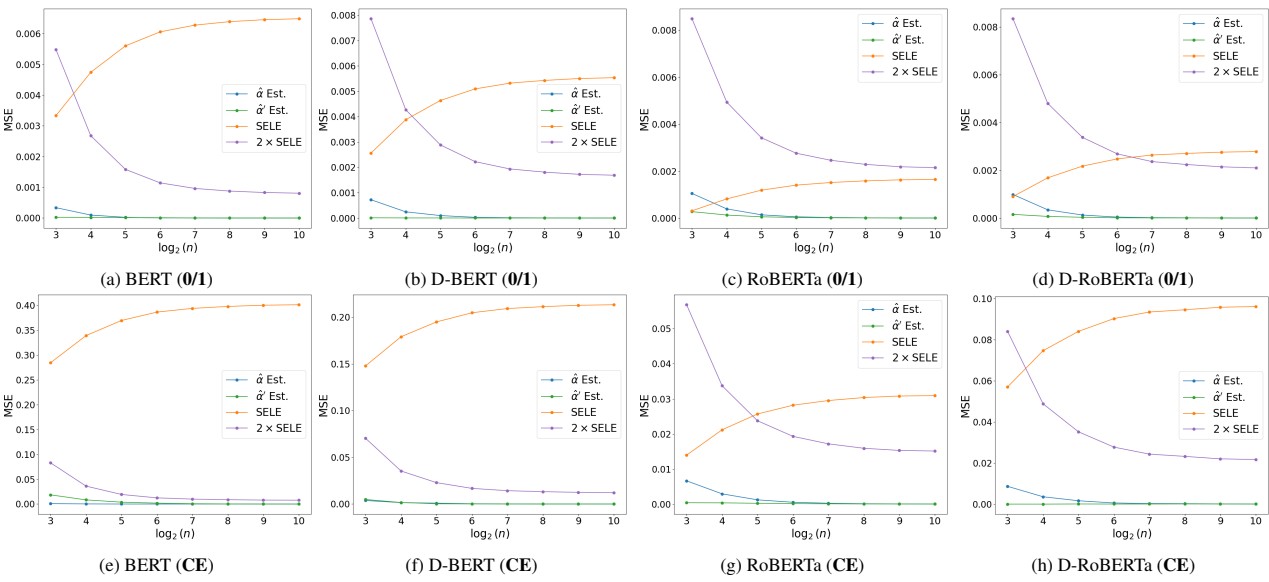

*Figure S24.* (**Amazon**) MSE of different finite sample estimators with **0/1** or **CE** loss. For each model architecture, we calculate the MSE of the estimators using a pre-trained model on batch samples derived from the test set.

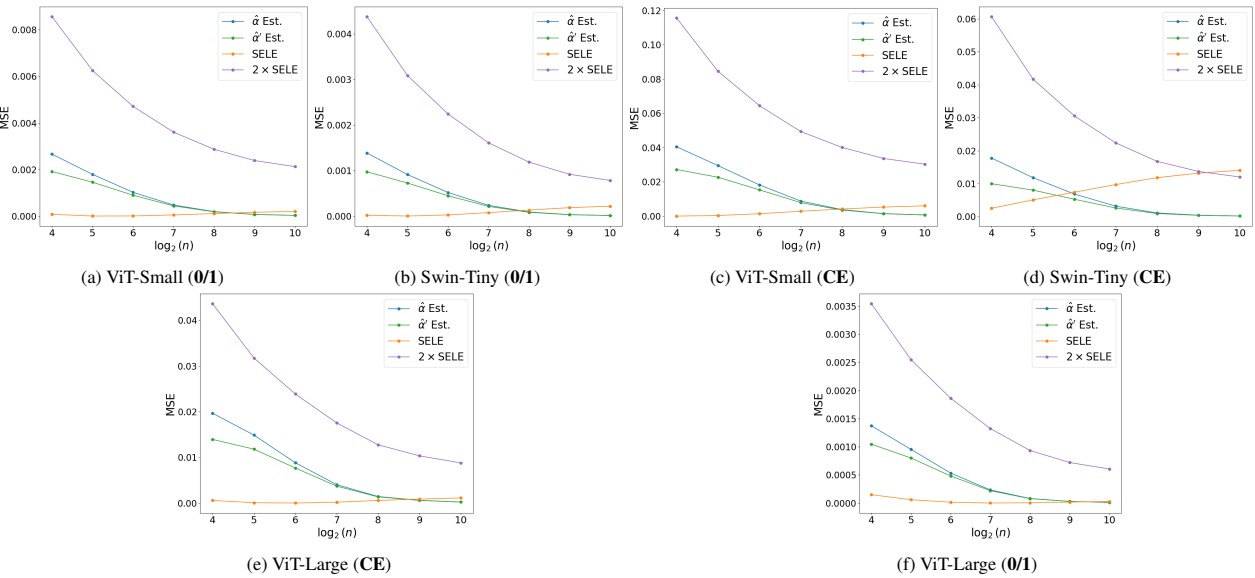

Figure S25. (**ImageNet**) MSE of finite sample estimators with **0/1** or **CE** loss. For each model architecture, we calculate the MSE of the estimators using a pre-trained model on batch samples derived from the test set.

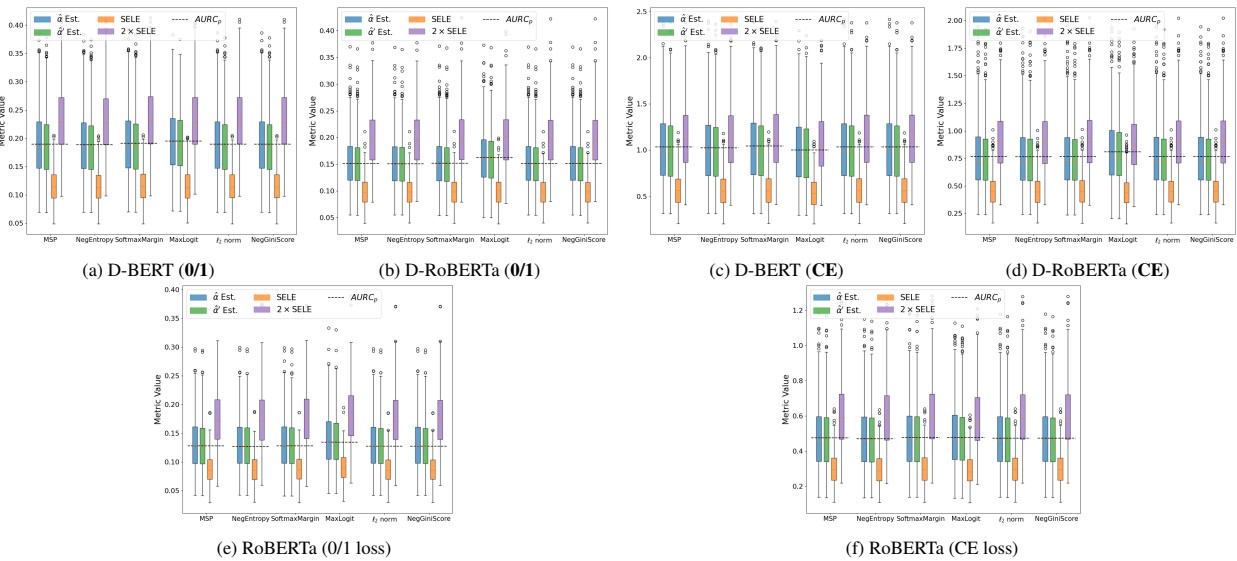

Figure S26. (**Amazon**) Finite sample estimators that utilize **0/1** or **CE** loss with different CSFs.

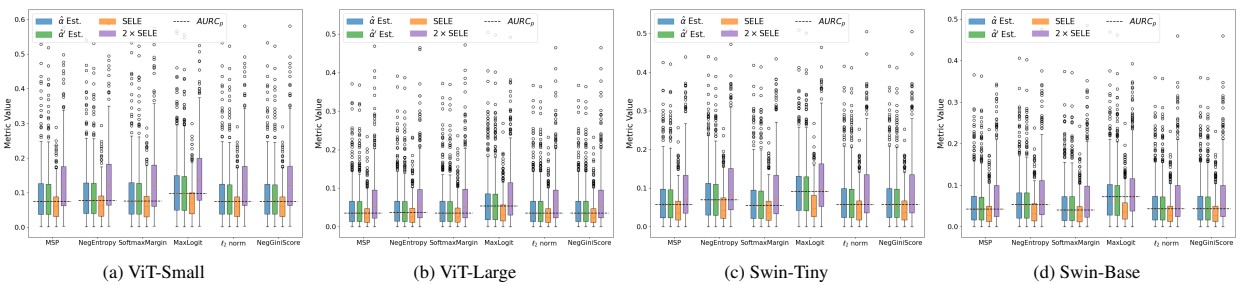

Figure S27. (**ImageNet**) Finite sample estimators that utilize **0/1** loss with different CSFs.

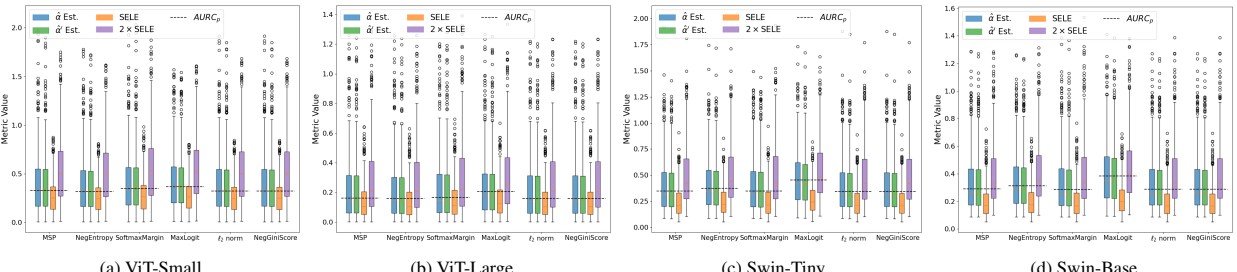

*Figure S28.* (**ImageNet**) Finite sample estimators that utilize **CE** loss with different CSFs.

