# OpenReview forum: "A Novel Characterization of the Population Area Under the Risk Coverage Curve (AURC) and Rates of Finite Sample Estimators"
_ICML.cc/2025/Conference — ICML 2025 poster_

### Official Review · Reviewer_BrTV · 2025-03-13

**Overall Recommendation:** 3

**Summary:**

This paper concerns evaluating the performance of *selective classifiers*, where the classifier has the option of abstaining from making a prediction when the confidence is low. For such classifiers, the Area Under the Risk Coverage curve (AURC) has been a commonly used (population) metric for evaluating their performances. This paper finds an equivalent representation of AURC, based on which an estimator is provided. Asymptotic properties of the estimator are analyzed and the proposed method is evaluated in simulations and real data examples.

**Claims And Evidence:**

The major claim of the paper is that it provides a new estimator for AURC that is consistent asymptotically and computationally efficient. The claim is supported by proofs and numerical evidence.

**Essential References Not Discussed:**

NA.

**Experimental Designs Or Analyses:**

Yes, I have checked the experimental design. My questions are:
1. What if one uses the estimator in Equation (10) or (11) directly (for example, in the numerical studies)? What is the advantage of the proposed method over it?
2. In some of the plots, the confidence intervals are quite wide and the performance of the candidate methods is not quite distinguishable. I wonder if more accurate results can be presented.
3. What is the take-away of Table 1? There do not seem to be significant differences across different methods.

**Methods And Evaluation Criteria:**

The estimator is constructed by first establishing an equivalent formulation of the population quantity and then plugging in sample-version estimates. I wonder, what if one uses the estimator in Equation (10) or (11) directly (for example, in the numerical studies)? What is the advantage of the proposed method over it?

**Other Comments Or Suggestions:**

NA.

**Other Strengths And Weaknesses:**

NA.

**Questions For Authors:**

1. What if one uses the estimator in Equation (10) or (11) directly (for example, in the numerical studies)? What is the advantage of the proposed method over it?
2. In some of the plots, the confidence intervals are quite wide and the performance of the candidate methods is not quite distinguishable. I wonder if more accurate results can be presented.
3. What is the take-away of Table 1? There do not seem to be significant differences across different methods.

**Relation To Broader Scientific Literature:**

This paper finds an equivalent representation of AURC, based on which an estimator is provided. Asymptotic properties of the estimator are established.

**Theoretical Claims:**

I briefly went over the outline of the proof for the consistency.

---

> ### Author Rebuttal · Authors · 2025-03-31
>
> ***Response to Reviewer BrTV:***
>
> **Q1:** See general response from "Notably, the Monte Carlo estimator using $\hat{\alpha}_{i}$...from a theoretical perspective."
>
> **Q2:** Take Fig. 3 as an example. When the confidence intervals are wide, it is typically due to the estimator being evaluated with a small sample size. This is expected, as small $n$ naturally leads to higher variability. In our analysis, we show that the convergence rate for our estimators is $\mathcal{O}(\sqrt{\ln(n)/n})$, which explains the wider intervals at smaller sample sizes. Nevertheless, as the sample size increases, the variance—and hence the confidence intervals—of the estimators converge as expected. With a sufficiently large sample size, both Monte Carlo estimators demonstrate consistency.
>
> **Q3:** In Table 1, we show that our estimators perform comparably to SELE and outperform standard cross-entropy risk minimization. While [1] describes the SELE score as a “computationally manageable proxy of AuRC”, we demonstrate that it is possible to directly optimize consistent estimators of AURC using our formulations—rather than relying on a lower bound, as SELE does. In general, optimizing a lower bound is often less effective than optimizing the true objective or an upper bound. We are not concerned by the fact that SELE achieves similar results in terms of optimization—indeed, optimization is not the primary contribution of this paper. Rather, our findings illustrate that population AURC can, in fact, be optimized via batch training using our estimators as loss functions. The main focus of this work lies in the estimation of AURC and the theoretical foundation that supports it—not in advocating for AURC optimization. Regarding SELE, although [1] claims that $2\times$SELE serves as an upper bound for the empirical AURC (Theorem 8), we show in Appendix Section A.2 that this is not the case. Furthermore, our empirical results indicate that SELE is not a consistent estimator of the population AURC.
>
> [1] Franc, Vojtech, Daniel Prusa, and Vaclav Voracek. "Optimal strategies for reject option classifiers." *Journal of Machine Learning Research* 24.11 (2023): 1-49.

---

### Official Review · Reviewer_3mMn · 2025-03-13

**Overall Recommendation:** 4

**Summary:**

This work addresses estimation of AURC risk aware classification by Selective Classifiers (SC).
It considers the theoretical properties of a pre existing AURC estimators, specifically SELE and proposes several new population based estimators with improved convergence properties.
Furthermore, they conduct empirical analysis of their findings to confirm the properties predicted by theory.

**Claims And Evidence:**

The claims are that the two proposed estimators have improved convergence properties compared to established approaches.
The claims are backed theoretically and largely check out empirically.

**Essential References Not Discussed:**

None to extent of my knowledge.

**Experimental Designs Or Analyses:**

The experiments were set up reasonably. Several datasets were used, effect of 0/1 as well as CE loss were investigated.
The proposed population estimators were also used to tune the models successfully.

**Methods And Evaluation Criteria:**

The proposed methods and evaluation criteria make sense.

**Other Comments Or Suggestions:**

Minor point: formulas in Proof 3.4 overflow into the page margins.

**Other Strengths And Weaknesses:**

The strength of the paper is in depth analysis of statistical properties of AURC estimators.

**Questions For Authors:**

1. The theoretical results derived apply to a fixed classifier. Can this approach be generalized to the epistemic risk with respect to multiple modes of the posterior distribution $p(\mathbf{w} \mid \mathcal{D})$ ?
2. How would introducing a cost function $c$ affect the properties of the estimators?

**Relation To Broader Scientific Literature:**

This paper falls into the broader work on risk aware machine learning.

**Theoretical Claims:**

The theoretical section is large, the proofs were superficially checked and appear to be in good order.

---

> ### Author Rebuttal · Authors · 2025-03-31
>
> ***Response to Reviewer 3mMn:***
>
> **Claims and Evidence:**
> We largely agree that our analysis focuses on the two proposed Monte Carlo estimators. In fact, one of them is equivalent to the widely used plug-in estimator, and our results confirm the effectiveness of this plug-in estimator from a theoretical perspective. Moreover, our analysis provides practical guidelines on which estimator to use and demonstrates that the novel estimator based on $\hat{\alpha}^{\prime}_i$ achieves the same convergence rate as the plug-in estimator.
>
> **Comments.**
> Thanks and we'll correct this format issue in the final version.
>
> **Question.**
>
> **Q1:** A straightforward way to address this issue is to consider the following expression:
>
> $$
> \mathbb{E}\_{f\sim \mathbb{P}(f|\mathcal{D})}[\text{AURC}(f)] = \mathbb{E}\_{f\sim \mathbb{P}(f|\mathcal{D})}\left[ \mathbb{E}\_{x} \left[ \mathcal{R}(f,g) \mid \tau = g(x)\right] \right]
> $$
>
> where $\tau$ is the threshold value. This can be computed directly using our method with Monte Carlo sampling. By applying Fubini’s Theorem, we can exchange the order of the two expectations, leading to:
>
> $$
> \mathbb{E}\_{f\sim \mathbb{P}(f|\mathcal{D})}[\text{AURC}(f)] = \mathbb{E}\_{x}\left[ \mathbb{E}\_{f\sim \mathbb{P}(f|\mathcal{D})} \left[ \mathcal{R}(f,g) \mid \tau = g(x)\right] \right].
> $$
>
> Since $g(x)$ depends on $f(x)$ under the posterior distribution $\mathbb{P}(f|\mathcal{D})$, and the predictions are made through a full Bayesian framework, this formulation allows the evaluation of AURC in a way analogous to the standard AURC for a fixed model. We can envision several ways to define potential quantities of interest based on model uncertainty. Many of these quantities could be potentially connected to AURC and epistemic risk, and exploring these relationships could open up valuable avenues for further investigation.
>
> **Q2:** Given an explicit cost function, it can naturally be incorporated into a loss function. If a cost is explicitly assigned to abstention, then evaluating AURC becomes unnecessary. Instead, with appropriate analysis—such as that in [2]—one can identify the optimal operating point that determines a threshold on the confidence score function (CSF). This is a well-studied area in the literature. However, it is outside the scope of this paper, as our focus is on analyzing the entire risk-coverage curve. Unlike approaches that rely on a fixed cost, AURC evaluates performance across all possible cost trade-offs, providing a broader view of model behavior under selective prediction.
>
> [2] Cortes, Corinna, Giulia DeSalvo, and Mehryar Mohri. "Learning with rejection." Algorithmic Learning Theory: 27th International Conference, ALT 2016, Bari, Italy, October 19-21, 2016, Proceedings 27. Springer International Publishing, 2016.

---

### Official Review · Reviewer_cuHV · 2025-03-13

**Overall Recommendation:** 2

**Summary:**

Introduces population Area Under the Risk-Coverage Curve (AURC) for selective classifiers. Statistical properties of these estimators are analyzed and evaluated on CIFAR datasets with VGG models.

**Claims And Evidence:**

Needs more evidence evaluating AURC optimization (convergence, computational difficulty, etc).




While mathematically sound, the practical benefits of the proposed approach over existing methods are not convincingly demonstrated through compelling real-world examples.

**Essential References Not Discussed:**

Missing discussion and comparion of previous work in learning with rejection:
* https://cs.nyu.edu/~mohri/pub/rej.pdf
* https://arxiv.org/html/2107.11277v3

**Experimental Designs Or Analyses:**

Improvements over SELE baselines are marginal. Analysis or discussion of computational overhead of the proposed estimators should be compared to baselines.

**Methods And Evaluation Criteria:**

Only one baseline (SELE) is compared. Other selective classification methods should also be evaluated as baselines.

**Other Comments Or Suggestions:**

How to generalize beyond classification accuracy. What about calibration, robustness to distribution shift, or performance on minority classes?

**Other Strengths And Weaknesses:**

Limited practical applicability and novelty. The novelty compared to existing work on AURC seems incremental.

**Questions For Authors:**

* What are the real-world implications of these marginal improvements?
* How do your estimators perform under distribution shift or with imbalanced datasets?

**Relation To Broader Scientific Literature:**

Connection to broader literature on uncertainty estimation in deep learning seems superficial

**Theoretical Claims:**

N/A

---

> ### Author Rebuttal · Authors · 2025-03-31
>
> ***Response to Reviewer cuHV: (continued): Please refer to the initial part of this response above.***
>
> **Essential Reference Not Discussed:**
> We agree that learning with rejection is related to AURC in relation to selective classifiers, where the model includes a reject option. However, this line of work is not directly relevant to the literature on AURC estimation, which is the primary focus of our study. As highlighted in [3], in the learning-with-rejection setting, the rejection threshold is learned by optimizing an explicit cost function (e.g., Eq. (9) in [3], Eq.(1) in [2]. This work primarily focuses on learning the rejector and evaluating models equipped with a rejector. In contrast, the objective of AURC is fundamentally different—it is an evaluation metric that measures the expected selective risk across all possible threshold values induced by the data. Optimizing AURC is equivalent to finding a model that achieves consistently better accuracy across the entire range of rejection thresholds. Although [3] uses the Area Under the Accuracy-Reject Curve (referred to as AURC in their Sec 3.2) to evaluate the trade-off between model accuracy and rejection, the intuition behind their Fig.7 aligns with the Area Under the Risk-Coverage Curve (AURC) used in our paper. However, they treat AURC purely as an evaluation metric and do not attempt to estimate or optimize it, which is the central goal of our work. While these papers provide interesting background on selective classification, they are not directly relevant to AURC estimation and analysis. Nevertheless, we will cite them in the final version in the introduction section as related work.
>
> **Weakness:**
> As noted in [3], AURC is an important evaluation metric for analyzing the risk-coverage trade-off. We believe it is valuable to understand the characteristics of this metric more deeply. In our work, we provide a novel definition of the population AURC (see Definition 3.2), framing it as a reweighted risk function, where the weighting term admits a closed-form expression (see Eq. (7)). To estimate the population AURC, we introduce two plug-in estimators derived via Monte Carlo methods. One of these corresponds to the widely used empirical AURC found in existing literature, while the other—based on the novel weight estimator $\hat{\alpha}^{\prime}_i$—is proposed in this paper. As highlighted in this paper, this novel formula achieves a computation cost $\mathcal{O}(n\log(n))$, offering a significant improvement over the naïve empirical AURC implementation. Importantly, we also establish convergence rates for both estimators, which, to our knowledge, have not been addressed in prior work. While [1] claims that “SELE loss is a close approximation of the AURC and, at the same time, amenable to optimization,” our results show that SELE is not a reliable estimator for the population AURC when compared with our estimators. Furthermore, our findings show that AURC can be effectively optimized via batch training using our estimators as loss functions, yielding significant improvements over standard cross-entropy risk minimization. We believe these results are important for the application of AURC estimation and optimization.
>
> **Other comments:**
> Thanks! These are indeed the directions we're considering for future work. However, our current goal is to first publish this paper, which provides a solid foundation for AURC estimation, independent of those other directions. AURC is an interesting metric in its own right and deserves a proper analysis—both in terms of its characteristics and how it should be estimated.
>
> **Q1:** These are not marginal improvements—they provide precise guidance on how AURC should be estimated and optimized. While SELE may achieve results comparable to our proposed estimator in terms of optimization, it is still not a reliable estimator of the population AURC. As we have emphasized, optimization is only a minor aspect of this paper, and we are comfortable with the fact that SELE performs similarly in that regard. Our primary focus is on the estimation side, for which we provide extensive theoretical and empirical analysis. In particular, our paper shows what convergence rates can be expected for these plug-in estimators, providing a theoretical underpinning for further analysis of the AURC objective.
>
> **Q2:** Estimation under distribution shift is an crucial and interesting direction, but it is beyond the scope of this paper. Our theoretical characterization of AURC and the convergence rates we establish provide a solid foundation for future work on understanding the effects of distribution shift and related challenges.
>
> **Reference**
>
> [1] Franc, V., Prusa, D., & Voracek, V. (2023). *Optimal strategies for reject option classifiers*. JMLR.
>
> [2] Cortes, C., DeSalvo, G., & Mohri, M. (2016). *Learning with rejection*. In ALT 2016.
>
> [3] Hendrickx, K., et al. (2024). *Machine learning with a reject option: A survey*. Machine Learning.

---

### Official Review · Reviewer_eX5V · 2025-03-14

**Overall Recommendation:** 4

**Summary:**

This paper proposes a new approach for learning classifiers for selective classification tasks where classifiers need not only make predictions, but also decide if it wants make a prediction or to abstain. It reformulates the area under the risk coverage curve (AURC) as an optimization objective during training of selective classifiers. Accompanying this, is an approach for estimating these classifiers using Monte Carlo methods The results show that the proposed approach outperforms the existing approach of performing uncertainty estimation on top of pre-trained models for selective classification.

**Claims And Evidence:**

The submission claims that Selective Classifier Learning (SELE) has some drawbacks – specifically, it learns to predict uncertainty scores based on outputs of pretrained models, and that the proposed approach of joint selective classification learning and uncertainty estimation can alleviate this issue. The results show that their framework outperforms SELE in terms of MAE.

**Essential References Not Discussed:**

The related works section details evaluation metrics used for selective classification, especially focusing on metrics related to the area under the risk coverage curve. This section also discusses approaches to uncertainty estimation and why the logit scaling over ensembling is chosen.

**Experimental Designs Or Analyses:**

The paper formulates a selective classifier based on the area under the risk coverage curve (AURC), and estimation via Monte Carlo methods. Instead of treating the AURC as a metric, the proposed approach suggests directly optimizing this metric during training.

**Methods And Evaluation Criteria:**

The risk-coverage curve is a standard tool for selective classification. When we need to abstain from predicting, coverage becomes an important metric of interest. Pairing coverage with performance on the test set in the form of risk, gives us the risk-coverage curve, the area under which is a suitable metric for evaluating selective classifiers.

**Other Comments Or Suggestions:**

N/A

**Other Strengths And Weaknesses:**

N/A

**Questions For Authors:**

N/A

**Relation To Broader Scientific Literature:**

Seamless integration between safety critical system and machine learning models depends on frameworks such as selective classification. Understanding (and estimating) the metrics that govern these types of frameworks is a step-forward towards deployment of machine learning models in real-world safety critical system.

**Theoretical Claims:**

I took a high-level look at the proofs, but they are beyond the area of my expertise. It would be best if someone with more expertise would kindly check them.

---

> ### Author Rebuttal · Authors · 2025-03-31
>
> ***General response:***
> We thank all reviewers for the valuable feedback. We hope our response has addressed all the concerns.
>
> **Supplementary Material:**
> We indeed included the appendix as supplementary material; however, while two reviewers did not see it, the other two did and were able to check it. The appendix contains extensive additional empirical results as well as detailed theoretical proofs.
>
> **Main Contribution:**
> The primary goal of this work is to characterize the population AURC metric, with an emphasis on its interpretation as a redistribution of the risk function. To study this, we employ Monte Carlo methods and introduce two estimators based on weights $\hat{\alpha}\_{i}$ and $\hat{\alpha}\_{i}^{\prime}$. Notably, the Monte Carlo estimator using $\hat{\alpha}\_{i}$ corresponds exactly to the commonly used empirical AURC. This is to say, the weight estimators in Eqs. (12), (13), and (17) are equivalent, leading to the empirical AURC formulation given in Eq. (10) or Eq. (11). Our statistical analysis of this estimator—including its bias, MSE, and convergence rate—shows that this estimator is, in fact, principled from a theoretical perspective. The estimator with $\hat{\alpha}_{i}^{\prime}$ is a novel contribution of this paper, and we establish a connection between the two estimators in Eq. (20). The statistical analysis of both estimators is new and, to our knowledge, has not been addressed in the existing literature.
>
> ***Response to Reviewer eX5V:***
> Thanks very much for your comments. We direct you to our general reviewer response, in particular the section about supplementary material.
>
> ***Response to Reviewer cuHV:***
>
> **About Claims and Evidence:**
> Although we show that the population AURC metric can be optimized using batch training, this is only a minor aspect of the paper and not its central focus. The core contribution of this work lies in characterizing the statistical properties of these two estimators. We not only establish their convergence rates theoretically in Proposition 3.6, but also empirically validate these results across multiple models (i.e. VGG13/16, ResNet 20/56/110/164, Bert, RoBERTa, ViT and Swin models) and diverse datasets (i.e. CIFAR10/100, ImageNet, Amazon).
>
> The reviewer states that "Needs more evidence evaluating AURC optimization (convergence, computational difficulty, etc.)" We wholeheartedly disagree. As shown in Figures 3–6 and Figures S9–S28 (in total **152** Figures), we have provided an overwhelming amount of evidence of the optimization and convergence of the estimators. The computational complexity is $\mathcal{O}(n\log n)$ due to the need to sort the points by the CSF, and requires only a linear number of forward passes. Theoretical convergence rates are rigorously proved and demonstrated empirically.
>
> **Experimental Designs Or Analyses:**
> In terms of AURC estimation, [1] proposed the SELE and claimed $2\times$SELE serves as an upper bound for the empirical AURC. Both SELE and its upper bound are used as baselines in our comparison. In terms of estimation performance, our two proposed plug-in estimators significantly outperform the SELE, which we show to be inconsistent. We demonstrate that they possess strong statistical properties—such as asymptotic unbiasedness and provable convergence rates—and validate these findings empirically through extensive experiments. These results provide strong evidence that our plug-in estimators are reliable for estimating the population AURC, whereas SELE and $2\times$SELE are not. In terms of AURC optimization, our estimators achieve performance comparable to the SELE score and both outperform models trained with the standard cross-entropy loss. While this optimization result is not the main focus of the paper, it serves to demonstrate that the population AURC can, in fact, be optimized using our estimators as loss functions within a batch training framework.
>
> **Relation To Broader Scientific Literature:**
> Our paper shows that AURC can be interpreted as a redistribution of the risk function, where the redistribution is guided by the confidence score function used to rank the dataset. This interpretation is the key characteristic of the population AURC. While we acknowledge that many other uncertainty estimation methods could potentially be integrated with AURC, this is not the main focus of our study. Instead, our work centers on AURC estimation and the statistical properties associated with it. Moreover, we empirically examine the AURC combined with uncertainty estimation methods such as MSP, Negative Entropy, MaxLogit, Softmax Margin, MaxLogit-$\ell_2$ norm, and Negative Gini Score. Our results, presented in Figure 6 and Figures S26–S28, show that the plug-in estimators consistently outperform SELE and $2\times$SELE, regardless of the chosen uncertainty estimation method. This is sufficient to demonstrate that SELE and $2\times$SELE are not reliable AURC estimators.

---

### Decision · Program_Chairs · 2025-05-01

**Decision:**

Accept (poster)

**Comment:**

In this paper, the authors study the Area Under the Risk-Coverage Curve (AURC), an important metric in the context of selective classification. They  present a formal statistical formulation of population AURC and derive empirical AURC plug-in estimators for finite sample scenarios using Monte Carlo methods. Statistical properties such as consistency are shown for these estimators, and convergence rates are analysed. Empirically, the effectiveness of the estimators is evaluated in several experimental studies.

Overall, this is a reasonably strong paper with a solid contribution to selective classification. The reviewers had some comments and made some minor suggestions for improvement, which the authors could address in their rebuttal.